# Liquid metal-tailored gluten network for protein-based e-skin

Bin Chen[1,2,8], Yudong Cao[1,2,8], Qiaoyu Li[2,8], Zhuo Yan[3], Rui Liu[4], Yunjiao Zhao[4], Xiang Zhang [5], Minying Wu[2], Yixiu Qin[3], Chang Sun[3], Wei Yao[1], Ziyi Cao[1,2], Pulickel M. Ajayan [5], Mason Oliver Lam Chee[6], Pei Dong[6], Zhaofen Li[7], Jianfeng Shen [1,9✉] & Mingxin Ye[1,9✉]

Designing electronic skin (e-skin) with proteins is a critical way to endow e-skin with biocompatibility, but engineering protein structures to achieve controllable mechanical properties and self-healing ability remains a challenge. Here, we develop a hybrid gluten network through the incorporation of a eutectic gallium indium alloy (EGaIn) to design a self-healable e-skin with improved mechanical properties. The intrinsic reversible disulfide bond/sulfhydryl group reconfiguration of gluten networks is explored as a driving force to introduce EGaIn as a chemical cross-linker, thus inducing secondary structure rearrangement of gluten to form additional β-sheets as physical cross-linkers. Remarkably, the obtained gluten-based material is self-healing, achieves synthetic material-like stretchability (>1600%) and possesses the ability to promote skin cell proliferation. The final e-skin is biocompatible and biodegradable and can sense strain changes from human motions of different scales. The protein network microregulation method paves the way for future skin-like protein-based e-skin.

[1] Institute of Special Materials and Technology, Fudan University, Shanghai, China. [2] Department of Chemistry, Fudan University, Shanghai, China. [3] State Key Laboratory of Molecular Engineering of Polymers, Department of Macromolecular Science, Fudan University, Shanghai, China. [4] State Key Laboratory of Food Nutrition and Safety, Tianjin University of Science & Technology, Tianjin, China. [5] Department of Materials Science and NanoEngineering, Rice University, Houston, TX, USA. [6] Department of Mechanical Engineering, George Mason University, Virginia, VA, USA. [7] RENISHAW (Shanghai) Trading CO.LTD, SPD, Shanghai, China. [8] These authors contributed equally: Bin Chen, Yudong Cao, Qiaoyu Li. [9] These authors jointly supervised: Mingxin Ye, Jianfeng Shen. ✉email: jfshen@fudan.edu.cn; mxye@fudan.edu.cn

The increasing demand for electronic skin (e-skin) in the fields of skin-attachable devices, robotics and prosthetics has motivated various cutting-edge technologies to endow e-skin with skin-like sensory capabilities and controllable mechanical properties, but unfortunately, e-skin with bio-compatibility presents great challenges for practical on-skin applications[1–3]. Therefore, despite the current existence of different kinds of reported synthetic materials, there is still a strong desire to explore biocompatible e-skin materials. Given that protein is a vital component of skin, proteins are the ideal option to provide e-skin with biocompatibility[4]. However, the design of e-skin with proteins is still in its infancy because precisely controlling the structure of proteins to obtain adjustable mechanical properties and self-healing abilities is fairly complicated.

Silk fibroin (SF), the dominant protein in e-skin research, has demonstrated feasibility for fabricating e-skins with tuneable mechanical behaviour and biocompatibility through complex plasticization or carbonization pretreatments[4–8]. In this regard, the gluten protein is proposed for preparing e-skin by a simple method. Upon hydration and kneading, gluten is known to form a cross-linked three-dimensional polymeric network through intra- and intermolecular covalent and noncovalent bonds[9–11]. The gluten network possesses various dynamic bonds, such as dynamic covalent disulfide (S-S) bonds and noncovalent H-bonds, thus guaranteeing the self-healing ability that most SF-based e-skins lack[1,11–14]. Generally, for e-skin preparation, the macroscopic mechanical performances of synthetic materials, such as supramolecular polymers, can be adjusted by constructing cross-linking positions inside the microscopic network structure[15–17]. However, enhancing the mechanical properties of the soft gluten network by achieving co-incorporation of physical and chemical cross-linking sites at the molecular level remains a challenge. Fabricating gluten-based e-skin through network regulation would lead to more sophisticated control over the mechanical properties and self-healing ability.

To address this issue, the abundant free sulfhydryl (-SH) groups in the gluten network can be explored to construct cross-linking sites since they can be used as ligands to form cross-linking bonds via metal-ligand coordinative interactions. Based on this idea, the eutectic gallium indium alloy (EGaIn), a promising candidate for improving the mechanical properties of soft materials, has the potential to be introduced into the gluten network due to its ability to interact with thiolate ligands[18,19]. Therefore, this work focuses on the structural characteristics of the gluten network to design an EGaIn/gluten-based e-skin (E-GES). Our strategy targets the dynamic -SH/S-S rearrangement mechanism in the gluten network[20] to introduce EGaIn as a chemical cross-linker and thus realize the establishment of hierarchical S-bonds in E-GES in which S-S bonds maintain the structural integrity[12] while EGaIn-SH coordinative bonds are supposed to dissipate energy[16]. Surprisingly, this structural adjustment strategy can induce changes in the gluten backbone conformation to obtain additional β-sheets as physical cross-linkers[21,22]. Thus, the high-density cross-linking sites and various dynamic bonds endow E-GES with exceptional mechanical strength and toughness, respectively. Moreover, E-GES can withstand spatial strain variations, behaving like conventional synthetic materials[23]. Finally, E-GES not only shows excellent cell biocompatibility in cytotoxicity tests but also promotes cell proliferation and slows cell apoptosis to a certain extent. Such characteristics make E-GES a competitive strain sensor for sensing strain signals from different human motions in daily on-skin applications.

In this work, targeting a current limited number of proteins and corresponding structural design strategy suitable for protein-based e-skin preparation, we successfully show that gluten is an attractive alternative to SF in the field of e-skin. Here, through the dynamic network microregulation mechanism of gluten networks, we realize a combination of liquid metal and protein to achieve gluten-based e-skin, based on whose dynamic bonds analysis, a reliable self-healing mechanism for fabricating self-healing protein-based e-skins is suggested. Besides, the gluten-based e-skin performs well in human motion strain sensing, comparable to synthetic material-based e-skins. This work can provide insights into metal-protein interactive mechanics for developing more proteins for designing human skin-like e-skins.

## Results

**Gluten network regulation by EGaIn.** Gluten extracted from wheat flour is a mixture of polymeric glutenins and monomeric gliadin subunits[24,25]. In the hydration and kneading process, glutenins tend to align and form a cross-linked three-dimensional gluten protein polymeric network through intra- and intermolecular S-S bonds and noncovalent bonds, thus contributing to the strength and elasticity, while gliadins combine with the formed glutenin structures via noncovalent bonds and influence the viscous properties of the protein matrix[10,26]. Notably, to produce a stronger protein network, a small amount of salt is needed during this process because the net positive charge of gluten can be shielded by salt, thus decreasing the electrostatic repulsion between gluten molecules and making them combine closely[27,28]. Different from wheat dough, there are no starch granules embedded in the framework of the E-GES network. As shown in Fig. 1a, E-GES was prepared by mixing gluten with EGaIn-dispersed solutions with the addition of a small amount of NaCl (2.8 wt.% E-GES) and kneading. According to the mass ratio of EGaIn in gluten, the obtained E-GES samples are named 1, 3, 5, and 10%, with the sample without EGaIn addition as a control denoted as 0%. The formation of different secondary structures of E-GES was revealed by analysing the amide I absorption bands in Fourier transform infrared (FTIR) spectra (discussed later) (Supplementary Fig. 1). The typical peak appearing at 521 cm$^{-1}$ in the Raman spectra confirms the formation of S-S bonds in E-GES, representing the trans-gauche-gauche conformation of S-S bonds, and the peak at approximately 2478 cm$^{-1}$ indicates the presence of -SH bonds (Supplementary Fig. 2)[29,30]. Furthermore, thiolated ligands have been demonstrated to be readily attached to the interface of EGaIn[18], so the abundant -SH groups of gluten are expected to provide reactive sites for EGaIn. This expectation is confirmed by the distinct binding energy shift in the S 2p region observed in X-ray photoelectron spectroscopy (XPS) spectra along with the redistribution of -SH/S-S contents in response to higher EGaIn concentration (Fig. 1b, c)[31]. The increasing trend of free-SH content, released from the combination with EGaIn during the analytical test, supports these observations. Furthermore, to ensure the integrity of the gluten network after introducing EGaIn, field emission scanning electron microscopy (FESEM) was used to observe the E-GES microstructure. Figure 1d, e show that the gluten network of E-GES is more regular and denser than that of the control sample, rather than loosely layered, meaning that the decrease in the S-S content does not influence the structural integrity of the gluten network. In addition, the energy-dispersive X-ray spectroscopy (EDS) mapping result reveals the presence of EGaIn in E-GES (Supplementary Fig. 3). Therefore, EGaIn has been successfully introduced into the gluten network through the construction of intermolecular EGaIn-SH coordinative bonds and in turn contributes to the adjustment of the gluten network. The obtained E-GES can be easily stretched by hand (Fig. 1f) and moulded into entirely different shapes (Fig. 1g), suggesting its ability to adhere well to irregular body surfaces.

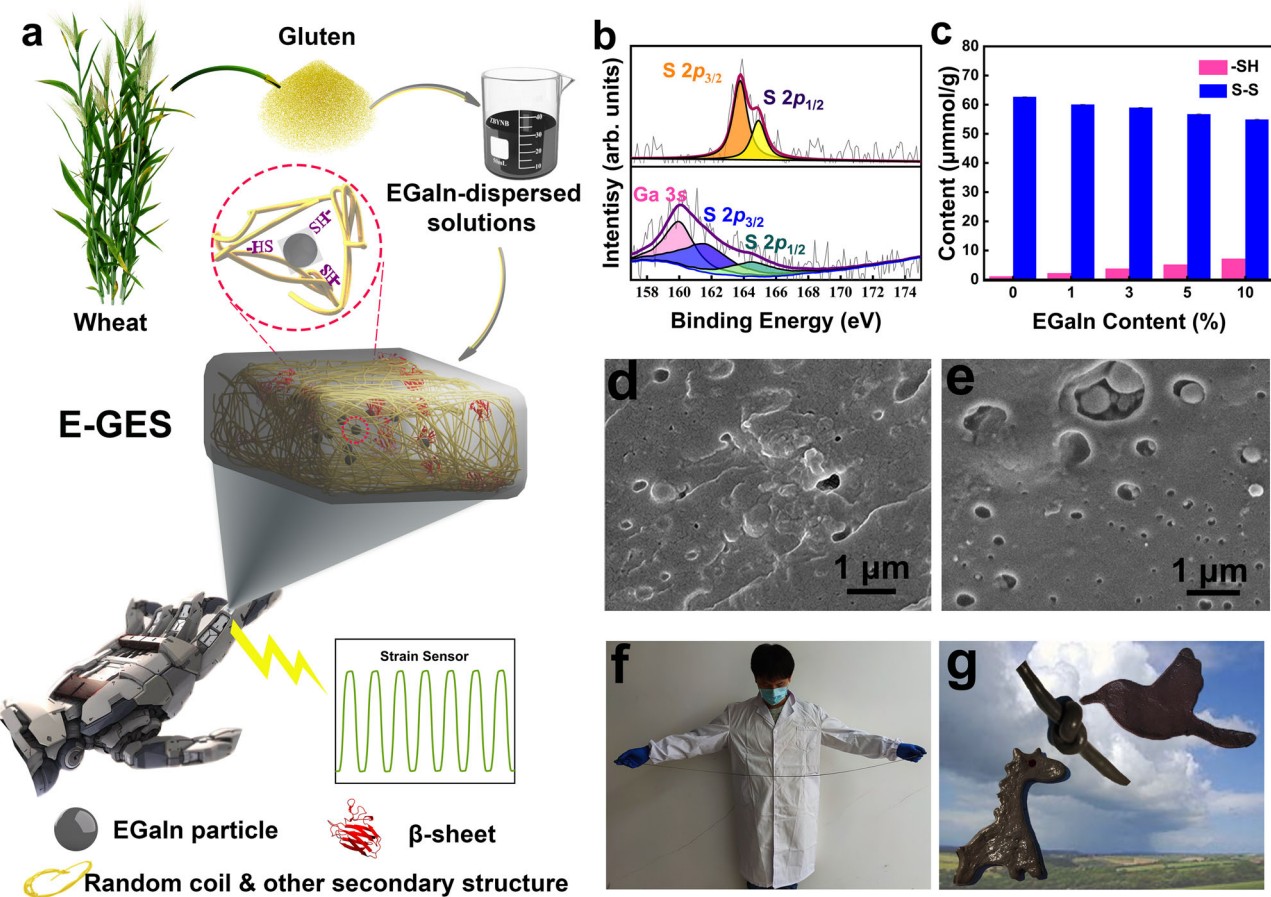

**Fig. 1 Schematic drawings of preparing E-GES and analysis of the interaction between EGaIn and -SH groups of gluten in E-GES. a** Schematic illustration of the E-GES fabrication process. **b** The XPS spectra of the control sample (top) and 5% E-GES (bottom). -SH groups are effective ligands for coordination with EGaIn, and S-Ga bonding can be revealed by XPS. For the control sample, the S 2$p$ region shows the S 2$p_{3/2}$ (orange) and S 2$p_{1/2}$ (yellow) components at binding energies of 163.8 eV and 164.9 eV, respectively, while these two peaks in the 5% E-GES sample show shifts in the low binding energy direction due to the change in electron density of S, appearing at 161.6 eV (blue) and 164.5 eV (green), respectively. This indicates the formation of EGaIn-SH coordinative bonds in E-GES, consistent with previous reports[16, 26]. Ga 3$s$ region is illustrated by pink colour. **c** Free -SH and S-S contents of different E-GES samples. Mean ± SD of three independent experiments were shown ($n = 3$). In the E-GES preparation process, with increasing amount of EGaIn, more -SH groups tend to interact with EGaIn to form coordinative bonds, thus reducing their ability to form S-S bonds with each other. Therefore, as the EGaIn content increases, the S-S content decreases, and a higher -SH content can be detected. **d, e** SEM micrographs of the control sample (**d**) and the 5% E-GES sample (**e**). Images shown are representative of three independent experiments (n = 3). Observations of the microstructure of the E-GES gluten network demonstrate that although the addition of EGaIn causes a decrease in the S-S content, the gluten network structure remains intact, meaning that the formation of EGaIn-SH coordinative bonds contributes to enhancing the gluten network. **f** Photograph of a stretched E-GES sample. The 5% E-GES sample can be easily stretched more than 10 times. **g** Photograph of E-GES with different shapes. E-GES can be moulded into different complex shapes, i.e., knot, bird and giraffe, which is beneficial to the design of different shapes for irregular human skin.

**Connections between the macroscopic behaviour and microscopic structure of E-GES.** The designed E-GES is considerably strengthened and toughened, as reflected in the increased tensile strength, tensile toughness and stiffness shown in Fig. 2a. An increase in the EGaIn content contributes to improved mechanical performance, with the maximum tensile stress, dissipated energy and Young's modulus increasing from 13.5 kPa, 228.2 kJ/m3 and 11.9 kPa to 51.2 kPa, 597.1 kJ/m3 and 25.2 kPa, respectively, when the EGaIn content is 5% (Supplementary Fig. 4). In contrast, the breaking strain exhibits a decreasing trend and shows a minimum value of approximately 1200%. For commonly commercialized e-skin substrates, such as poly(dimethylsiloxane) (PDMS) (Sylgard 184), polyurethane (SG80A) and styrene-ethylene-butadiene-styrene (SEBS), their stretchability is lower than that of E-GES, exhibiting breaking strains of 200%, 700% and 280%, respectively, but they are stiffer than E-GES, with Young's moduli of 0.4 MPa, 1.73 MPa and 3.83 MPa,

respectively[15]. However, the Young's moduli of these commercial synthetic materials are often greater than the maximum value of the skin's Young's modulus (600 kPa, the mechanical properties of skin in different parts of the body differ)[32]. Compared with these materials, the softer and more stretchable E-GES may become a better choice for preparing e-skin. In daily human motions, human skin generally suffers from approximately 30% tensile strain, and some reports suggest that skin-like wearable flexible hybrid electronics with stretchability as high as 100% favour the acquisition of high-quality signals from skin[1,32]. Thus, the trade-off of the tensile strain of E-GES is acceptable and quite normal, and its remaining stretchability is comparable to that of many synthetic materials, a rarely reported phenomenon for protein-based materials in previous studies (Fig. 2c)[15,17,23,33–35]. Encouragingly, compared with the low stretchability of currently established protein-based e-skin, the high stretchability of E-GES not only makes it suitable for on-skin sensing applications but

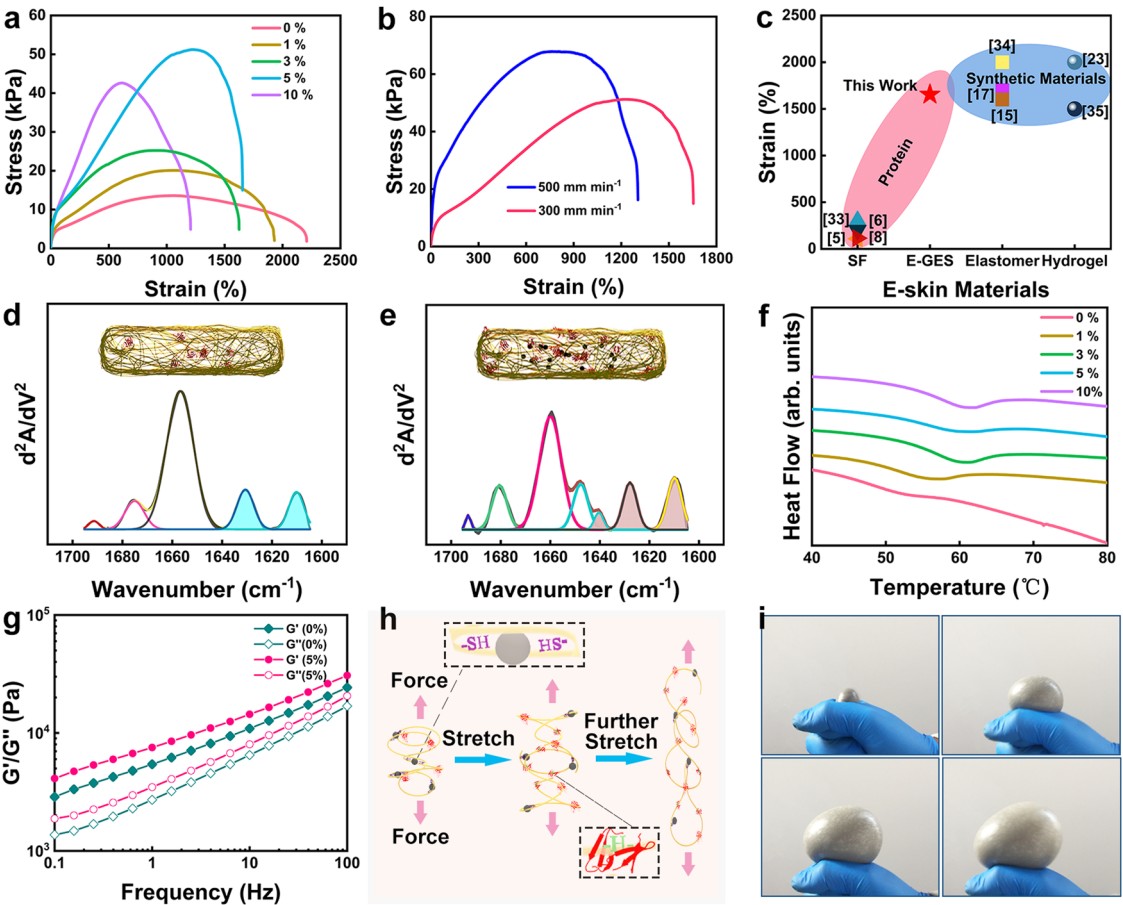

**Fig. 2 Characterization of the macroscopic mechanical properties and microscopic structures of E-GES. a** Stress-strain curves of different E-GES samples. **b** Stress-strain curves of 5% E-GES at different strain rates. The E-GES sample can resist the increasing strain rate without showing brittle fracture and exhibits a rising tensile toughness, meaning the breakage of more dynamic chemical bonds in the tough E-GES network to dissipate energy when a high tensile rate is applied. **c** Maximum tensile strain comparison between protein-based e-skins and synthetic material-based e-skins. **d, e** Calculated secondary structure results of the control sample (**d**) and 5% E-GES (**e**) from the analysis of the amide I band in their FTIR spectra (Supplementary Fig. 1). The peaks within the range of 1610-1640 cm$^{-1}$ are attributed to β-sheets, filled with blue (**d**) and brown (**e**) colour, respectively. Inset: Content of β-sheets and EGaIn in the E-GES network. **f** DSC thermograms of different E-GES samples. **g** Frequency sweep curves of $G'$ and $G''$ for the control sample and 5% E-GES. **h** Schematic illustration of the stretching process of E-GES. The presence of EGaIn-SH coordinative bonds and the intermolecular H-bonds of β-sheets favour energy dissipation. **i** Photographs of the inflation process of E-GES. The E-GES sample can be sealed on the inflation port (port diameter of 1 cm) of an air pump and then inflated into a large balloon.

also endows it with potential for usage in robotics, as a highly stretchable e-skin will enable a robot to assume various shapes and achieve high degrees of freedom in movement[1]. Importantly, the preparation of gluten-based e-skin is easier than that of the commonly established SF-based e-skin, which can be reflected in the following aspects: (i) extracting gluten from wheat is easier than preparing SF solutions from *Bombyx mori* cocoon fibres[36]; (ii) the obtained gluten is more stable and easier to store than SF solutions; and (iii) due to the mismatch of mechanical properties between SF and the skin, many complicated strategies, such as plasticization or carbonization, are required to adjust the mechanical properties of SF to prepare SF-based e-skins[5,7], whereas the strategy for the design of gluten-based e-skin is far simpler and more convenient. Therefore, gluten can be used as a new attractive protein material for e-skin preparation.

Insights into the microscopic structure of E-GES were obtained through FTIR analysis. The changes in the gluten backbone conformation were revealed by analysing amide I bands (1700–1600 cm$^{-1}$), and the deconvolution of amide I bands can be used to qualitatively and quantitatively evaluate gluten secondary structures, as illustrated in Fig. 2d, e and Supplementary Fig. 5[12,13,37].

According to previous studies, the change in α-helix content can indicate the integrity of the protein network because an increasing content is related to the construction of a more ordered protein network structure, while a decreasing content suggest the breakage of intermolecular S-S bonds and rearrangement of protein molecules[38,39]. The addition of EGaIn causes the α-helix content in E-GES to decrease, which may result from the rearrangement of -SH/S-S induced by EGaIn and thus the reduction of intermolecular S-S bonds; however, all E-GES samples have more than 33% α-helix structures in their gluten network (Supplementary Fig. 6). The α-helix contents were approximately 20% to 30% in previous gluten modification research, so E-GES actually has a more ordered gluten network structure[12,13]. In addition, the β-sheet is referred to as the most stable conformation of gluten, and its content is positively related to the viscoelasticity and rigidity of the gluten network[12]. The incorporation of EGaIn facilitates the formation of a higher proportion of β-sheets at the expense of α-helices in a dose-dependent manner. This result is likely attributed to the reduction of intermolecular S-S bonds and the promotion of interactions among intramolecular H-bonds induced by EGaIn[12], so EGaIn

facilitates the construction of a more rigid and highly viscoelastic gluten network in E-GES. Meanwhile, notably, these nanocrystal-like β-sheets can act as many physical cross-linking points embedded in the amorphous protein network, thus defining the mechanical properties of the protein, as widely found in studies on silk protein[21,40]. Generally, when protein chains are cross-linked, their mobility is limited, causing an increase in the denaturation temperature ($Tp$) of the protein. Thus, $Tp$ is an important indicator of changes in protein chains[22,41]. Differential scanning calorimetry (DSC) was used to detect the $Tp$ of different E-GES samples (Fig. 2f). $Tp$ increases from 48.26 °C to 58.17 °C under different EGaIn dosages, indicating that the mobility of gluten protein chains is restricted with the addition of EGaIn, and thus, the gluten network spatial structure becomes stronger and more ordered, instead of tending to becoming denatured and weaker (Supplementary Table 1)[38]. Moreover, in addition to the β-sheet physical cross-linking sites, there are abundant chemical cross-linking points because of the coordinative interactions between EGaIn and the -SH groups of gluten, so the adjustment of the E-GES gluten network should be attributed to the synergistic promotion of these two cross-linking mechanisms[22]. The changes in the E-GES networks can also be observed from the variation in their viscoelasticity[42]. The rheological characterizations show that all samples display a solid-like behaviour with a storage modulus ($G'$) greater than the loss modulus ($G''$), and E-GES samples have $G'$ and $G''$ increased by one to two times compared with the control sample (Fig. 2g and Supplementary Fig. 7). The enhanced viscoelasticity of E-GES agrees well with its enhanced mechanical performance in tensile tests. These outcomes indicate that we successfully created a hybrid cross-linking gluten network in E-GES through the construction of additional coordinative bonds by EGaIn and the conformational transition to the β-sheet induced by H-bond rearrangement[16,43]. In general, the improvement in the macroscopic strength and toughness of E-GES can be explained by the structural adjustment according to the following proposed mechanism: The presence of two different cross-linkers endowing E-GES with various dynamic chemical bonds leads to highly folded gluten protein chains, thus developing a hierarchical network structure, which is beneficial for withstanding tensile changes and dissipating energy (Fig. 2h). This mechanism is further corroborated by the increases in the Young's modulus and dissipated energy with increasing strain rates (Fig. 2b) and the gradually growing internal fracture of the E-GES network in the loading-unloading cycle tests with varying maximum strains (Supplementary Fig. 8), demonstrating the stiffness and toughness of the E-GES network structure. In addition, the robust E-GES network can withstand three-dimensional spatial strain variations, which is easily visualized through a simple and intuitive mechanical experiment. As shown in Fig. 2i, the membrane-shaped E-GES can be inflated into a balloon with a size range several orders of magnitude larger than before, which is an amazing phenomenon for protein-based materials or hydrogel-like materials.

**Self-healing ability and strain sensitivity of E-GES.** As is widely acknowledged, when the wheat dough is torn, dough fragments can be rolled up and kneaded, thus forming a reshaped dough without obvious performance degradation[11]. The principle behind this is the dynamic construction of and conversion between S-S bonds and free -SH groups with the assistance of intramolecular and intermolecular H-bonds involved in the gluten network rebuilding process. Naturally, this intrinsic characteristic endows gluten networks with remarkable self-healing ability for fabricating e-skin. The self-healing efficiencies of E-GES were evaluated through the following two aspects: stiffness

recovery (Young's modulus-based) and toughness recovery (dissipated energy-based). The stress-strain curves demonstrate that the healed E-GES recovers 97.9% of its stiffness and 51.4% of its toughness after healing from complete cutting (Fig. 3a). To examine the ultimate self-healing property of E-GES during the tensile experiment, two cut-off E-GES pieces were combined for 1 min without any further treatment; under this seriously restricted healing condition, the contact interfaces cannot fully disappear in the healed E-GES. This visible defect results in a local stress concentration, which is the main reason why the toughness decreases. However, the remaining stretchability was still measured to reach 700%, along with an almost unchanged Young's modulus, suggesting the exceptional self-healing ability of the strong E-GES network.

The self-healing mechanism based on the reconstruction of dynamic bonds is widely accepted to be more preferable for e-skin applications[1]. After introducing EGaIn into the gluten network to construct cross-linking bonds, there are three kinds of dynamic chemical bonds in E-GES, namely, S-S bonds, H-bonds and EGaIn-SH coordinative bonds. The self-healing mechanism of gluten networks based on reformation of dynamic S-S bonds and H-bonds has been accepted and reported in dough-based flexible electronics[11,14], and liquid metal-sulfide interactions have also been proven to be effective in the self-healing process of a liquid metal-embedded sulfur polymer[44]. Therefore, the self-healing ability of E-GES can be inferred to mainly depend on the influence of the introduced EGaIn on S-S bonds and H-bonds in the gluten network. How EGaIn influences S-S bonds in E-GES has been discussed above based on the dynamic -SH/S-S rearrangement mechanism in the gluten network, so information on H-bonds in the E-GES network needs to be investigated. In the established theories about the gluten network, tyrosine (Tyr) residues that periodically appear in the gluten proteins are involved in the construction of the gluten network by forming H-bonds, so information about H-bonds can be acquired by analysing the microenvironment of Tyr residues[29,45,46]. Generally, the Tyr doublet in Raman spectra, arising from the Fermi resonance of the ring-breathing vibration and an overtone of an out-of-plane ring-bending vibration of the para-substituted benzene ring, is mostly used to analyse the state of Tyr residues (Fig. 3b)[47] because the variation in the intensity ratio of the Tyr doublet ($I_{861}/I_{837}$) provides H-bonding information of Tyr residues[29,48]. An increase in the $I_{861}/I_{837}$ ratio indicates that Tyr residues tend to be exposed to a hydrophilic environment and possess the ability to form strong hydrogen bonds, serving as proton acceptors, while a decreasing value means that Tyr residues are in a buried state. The ratio changing from 1.17 to 1.65 shows that the combination with EGaIn promotes Tyr residue exposure (Supplementary Fig. 9)[49], meaning that the exposed Tyr residues contribute to the formation of hydrogen bonds as proton acceptors or can interact with each other to form intermolecular hydrogen bonds (Supplementary Fig. 10)[46,47].

Hence, based on the observations of Tyr residues of gluten chains and the presence of different dynamic chemical bonds in the gluten network, the proposed self-healing mechanism of E-GES can be summarized as follows: the introduction of EGaIn induces the formation of EGaIn-SH coordinative bonds and improves the ability of Tyr residues to form hydrogen bonds in the gluten network. Based on the existence of these two dynamic chemical bonds and the inherent disulfide bonds, when two cut pieces of E-GES contact each other, the dynamic reconstruction of these three dynamic chemical bonds in the contacting interfaces can be speculated to facilitates the initial self-healing process. Meanwhile, the disulfide dynamic covalent exchange continually promotes the eventual recovery process, and the broken gluten network is finally rebuilt through the formation of cross-linked S-S bonds and intra- and inter-molecular H-bonds (Fig. 3c). Similarly, a self-healing mechanism based on the combination of S-S bonds, H-bonds and metal-ligand

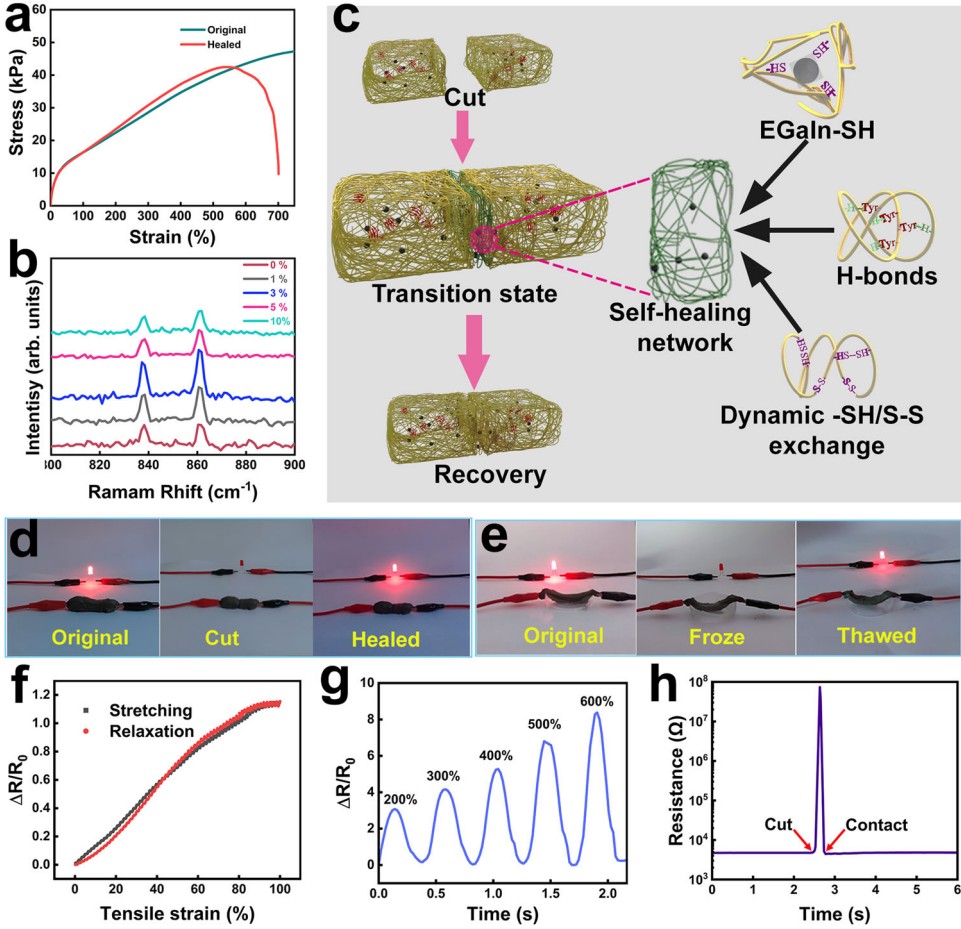

**Fig. 3 Characterization of the self-healing ability and strain sensitivity of E-GES. a** Stress-strain curves of pristine (green) and self-healed (red) 5% E-GES samples. In the self-healing process, two fresh cut-off E-GES pieces are simply placed in contact with each other without using other assistance methods, and the reshaped E-GES almost recovers its original Young's modulus in 1 min. **b** Raman spectra of different E-GES samples. The Tyr doublet peaks appear at 861 cm$^{-1}$ and 837 cm$^{-1}$. **c** Schematic illustration of the proposed self-healing mechanism of the E-GES network. The green E-GES network represents a self-healing transition state, arising from the formation of various dynamic chemical bonds, including EGaIn-SH coordinative bonds, noncovalent H-bonds and dynamic S-S covalent bonds, in the contacting cut interfaces. **d**, **e** Photographs of E-GES connected to an LED bulb in different situations. There is liquid nitrogen in the plastic container (**e**). **f** Stretching-releasing cycle with a strain of 100%. **g** Successive stretching-releasing cycles with strain increasing from 200% to 600%. **h** Resistance changes of E-GES after healing from complete cutting.

bonds has also been reported in a thioctic acid-based supramolecular polymer[16]. Obviously, the above discussion has successfully shown that the self-healing principle based on the reconstruction of dynamic bonds also applies to protein-based e-skin, suggesting that exploring the dynamic covalent bonding of proteins or constructing additional dynamic bonds inside protein structure is a great strategy to endow protein-based e-skins with self-healing ability. Furthermore, the excellent self-healing ability of E-GES is also reflected in the conductivity recovery, with lighting of an LED bulb with slight contact between two cut interfaces (Fig. 3d). Intriguingly, we find that ultralow temperatures cannot cause irreversible damage to E-GES because it can recover its mechanical and conductive properties to a great extent as the temperature rises (Fig. 3e and Supplementary Fig. 11). Therefore, in addition to mechanical damage, low- temperature damage cannot cause permanent damage to the recoverability of E-GES at room temperature.

The electrolyte salt NaCl was added to E-GES during the preparation process, so it can serve as an ion carrier, thus endowing E-GES with the ability to be used as a strain sensor, relying on the resistance changes to detect variations in applied strains (Supplementary Fig. 12). This sensing mechanism has been proven to be effective in the gluten network in dough-based e-skin[11]. The strain sensitivity of E-GES was systematically

evaluated based on the resistance changes caused by different stretching-releasing experiments. As shown in Fig. 3f, the resistance of E-GES first increases when stretched and then decreases after release, and no obvious conductivity hysteresis is observed, indicating the electrical stability as a consequence of the elasticity of the polymer network. Furthermore, in successive stretching-releasing cycles, the increasing resistance well matches the gradually increasing strain (Fig. 3g). Moreover, after healing from complete cutting, the healed E-GES is capable of restoring its original resistance (Fig. 3h). Similar to the performance of the original sample in the repeated stretching-releasing cycles, the signal changes of the healed E-GES also remain stable (Supplementary Fig. 13). The above experiments demonstrate that the electrical signal changes of E-GES are closely related to the applied strain variations, meaning that E-GES possesses excellent strain sensitivity and is well qualified for fabricating strain sensors. Notably, hydrogel-like E-GES is more convenient to prepare strain sensors, with neither the introduction of elastomeric polymer substrates nor the complex preliminary treatment, compared with reported SF-based e-skins[5,7].

In addition, e-skins are inevitably exposed to sweat in daily use, so the performances of E-GES under the impact of artificial sweat was investigated. In the self-healing test, the cut E-GES was

daubed with artificial sweat, including the cut interface, to explore its self-healing ability. The stress-strain curves show that under the impact of artificial sweat, the healed E-GES recovers 70.9% of its stiffness and 39.5% of its toughness after healing from complete cutting (Supplementary Fig. 14a). Compared with the results of the self-healing tests without the introduction of artificial sweat, the self-healing ability of E-GES clearly decreases. The reason for this is that the artificial sweat contains different salts, such as $Na_2SO_4$, $Na_3PO_4$, and $Na_2HPO_4$. According to the studies on the effects of the Hofmeister salt series on gluten network formation, different cations and anions have different influences on gluten network formation[27,28], so the existence of different salts in artificial sweat will influence the reconstruction of the E-GES network and cause a decrease in its self-healing ability. E-GES can maintain its strain-sensing abilities well under sweat. Supplementary Fig. 14b shows that the resistance of E-GES increases during stretching and then decreases during release, and no obvious conductivity hysteresis is observed in this process; in the repeated stretching-releasing cycles, the signal changes of E-GES also remain stable (Supplementary Fig. 14c). Therefore, when E-GES is exposed to sweat, its strain-sensing ability can be inferred to be unaffected. Moreover, even if E-GES is mostly immersed in artificial sweat, the LED bulb keeps lighting up, suggesting good electrical conductivity of E-GES in artificial sweat (Supplementary Fig. 14d). Hence, E-GES has the potential to sense the strain changes from human motions in daily use.

**Biocompatibility of E-GES**. Apart from the necessary stretchability and self-healing ability, qualified e-skin should be biocompatible. To explore the biocompatibility of E-GES, first, cytotoxicity tests against human epidermal keratinocyte cell lines (HACAT cells) and human dermal fibroblast cell lines (HSF cells) were carried out to evaluate the cytocompatibility of E-GES, and the overall assessment was performed by analysing the cell proliferation, viability and apoptosis. As shown in Fig. 4a, b, the cell proliferation in the experiment groups increases with increasing E-GES concentration. Intriguingly, the results of the experimental groups are better than those of the control group, indicating that the addition of E-GES can improve the proliferation of the two selected cells. This is mainly due to the presence of cysteine residues in E-GES, meaning that E-GES shows good cell compatibility and benefits cell proliferation. Regarding the cell viability, nearly all cells (HACAT and HSF cells) in the experimental groups (cultured with 10 mg/mL E-GES leaching solution) were alive (green fluorescence) (Fig. 4c, d), similar to the control group, with good biocompatibility (Supplementary Fig. 15a, b). In addition, the number of dead cells showed no significant difference between the experimental and control groups, which indicated that the cells grew well in the E-GES leaching solution. The apoptosis of cells at different stages was studied. From Fig. 4e, f, the proportion of normal living cells is more than 95% (lower left quadrant). Early apoptotic cells (lower right quadrant), late apoptotic cells (upper left quadrant), and necrotic and late apoptotic cells (upper right quadrant) are present, but their proportions are small. There is no significant difference between the experimental and control groups (Supplementary Fig. 15c, d), indicating that E-GES can promote cell proliferation and slow cell apoptosis to a certain extent. The above results powerfully demonstrate that E-GES possesses good cytocompatibility.

Furthermore, we specifically designed different experiments to estimate the effect of E-GES on rabbit skin, as shown in Fig. 4g, h. For skin with an artificial #-shaped wound, the attachment of E-GES does not affect wound healing, and newborn rabbit hair can be observed on the wound after seven-day attachment experiments, similar to the result of the control wound without

any treatments (Supplementary Fig. 16). In addition, no adverse reactions are observed on the surrounding healthy skin. Given that e-skin inevitably reacts with sweat in daily use, whether sweat will be polluted by e-skin and influence the health of the skin is another important problem. With this in mind, E-GES was totally immersed into artificial sweat for 24 h, and the treated sweat was applied to healthy skin and the skin with a #-shaped wound (Fig. 4h and Supplementary Fig. 17). Observation of the skin over a seven-day period definitively proves the excellent safety of E-GES when in contact with sweat. To protect the environment, we believe that an attractive biocompatible e-skin candidate should also be biodegradable. The disappearance of E-GES in a pepsin solution and the appearance of mould on the surface of E-GES in moist soil illustrates its eco-friendly biodegradability (Supplementary Fig. 18)[6,14].

**E-GES based strain sensor**. The strain-sensing performances of E-GES was verified by attaching it to different body parts (Fig. 5a). As shown in Fig. 5b, the bending degree of the forefinger is closely associated with the variations in the resistance, and an increase in the bending angle leads to an increase in the resistance, which can return to the original value when the forefinger is straightened. Rotation of the wrist by a fixed angle generates clear and repeatable resistance signals (Fig. 5c), and large movements of the knee can also be detected by E-GES (Fig. 5d). Furthermore, to test the strain sensitivity of E-GES to its limits, we examined its ability to respond to the thrust arising from breathing. E-GES embedded in a mask can continuously monitor the breathing motion of the volunteer, demonstrating its potential application for real-time health monitoring (Fig. 5e). The slight strain change from blinking can also be detected by E-GES when it is attached to the middle of the forehead (Fig. 5f). Therefore, compared with the established SF-based e-skins[6,7,50], E-GES can not only sense strain changes from human motions of different scales but also be used as a built-in sensor integrated in a mask to detect the slight changes induced by breathing, suggesting that E-GES is an attractive e-skin for strain sensing and that gluten is another potential protein for e-skin preparation in addition to SF. Moreover, because of its improved gluten network, E-GES can transform the volume change of a balloon into the variation in resistance signals by attaching E-GES to the surface of the balloon. Figure 5g, h show the increasing and repeatable signals corresponding to the gradual inflation process and the cyclic inflation-deflation process, respectively. This means that the robust E-GES network exhibits superior sensing abilities even after suffering from three-dimensional spatial changes in strain, a phenomenon currently rarely reported to the best of our knowledge. Therefore, in addition to sensing strain signals of human motions, E-GES possesses the potential to be used in robotics, in which e-skin with high stretchability in three dimensions will enable robots to perform more complicated motions, rather than simply imitating human motions[1]. In addition, E-GES was further examined by comparing the performance of an E-GES sample with a notch and that of a normal E-GES sample (Supplementary Fig. 19). Finally, cyclic stretching-releasing sensing experiments demonstrate the stable and recoverable sensing ability of E-GES (Fig. 5i), which results from the good recoverability of the E-GES network (Supplementary Fig. 20). Hence, E-GES is robust and has the ability to resist mechanical damage in daily use, comparable to the robustness of e-skins prepared from synthetic materials, such as PDMS and poly (acrylic acid)[35,51]. From the above experiment results, gluten protein can be inferred to have the potential to be an attractive alternative to synthetic materials for designing e-skin with more skin-like strain sensitivity or to be a new elastic polymer substrate

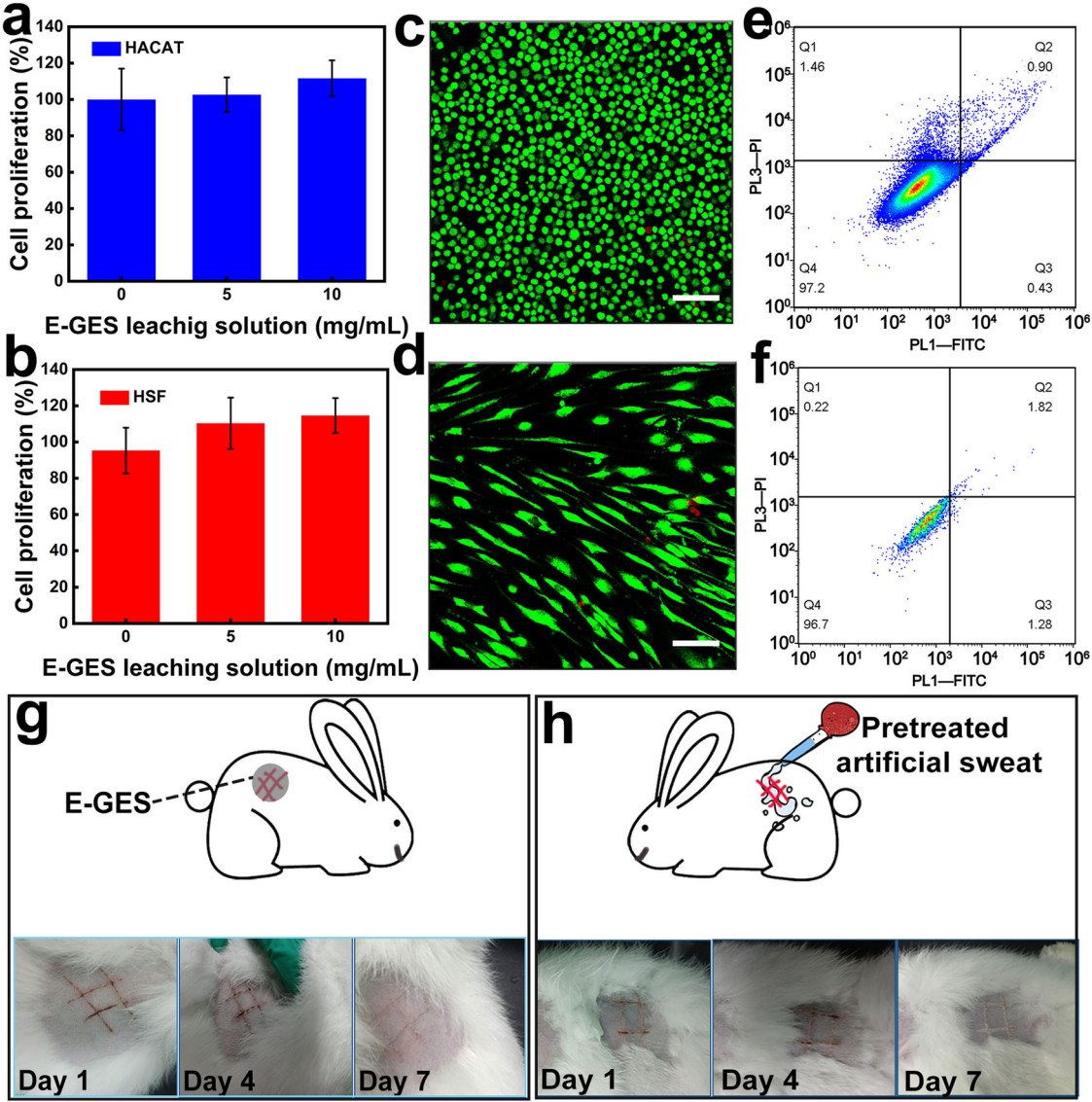

**Fig. 4 Biocompatibility tests of E-GES. a, b** Cell proliferation tests of HACAT (**a**) and HSF (**b**) cells in different concentrations of E-GES leaching solutions. Mean ± SD of three independent experiments were shown ($n = 3$). **c, d** Viability of HACAT (**c**) and HSF (**d**) cells cultured with 10 mg/mL E-GES leaching solution. Live cells and dead cells are illustrated by green colour and red colour, respectively. Scale bar=100 μm. Images shown are representative of three independent experiments ($n = 3$). **e, f** Apoptosis tests of HACAT (**e**) and HSF (**f**) cells cultured with 10 mg/mL E-GES leaching solution. Images shown are representative of three independent experiments ($n = 3$). **g, h**, Photographs of biocompatibility experiments of E-GES on rabbit skin. The skin was treated with E-GES (**g**) and pretreated artificial sweat (**h**). The pretreated artificial sweat was prepared by immersing E-GES into the artificial sweat for 24 h. Images shown are representative of four independent experiments ($n = 4$).

in the field of e-skin. However, the e-skins designed with synthetic materials or SF have achieved integration of different sensing abilities. For example, PDMS-based e-skin has been reported to have both strain and electrocardiogram sensing capabilities at the same time[51], and SF-based e-skin can be used as humidity or temperature sensors[8]. In contrast, the sensing ability of E-GES is relatively single function and needs to be further developed. Therefore, utilizing gluten to design e-skins with multifunctional sensing abilities is still a challenging task. Achieving regulation of tertiary and quaternary structures of gluten may be a key point in developing various sensing abilities of gluten-based e-skins.

## Discussion
In summary, we report a simple strategy to design gluten protein for fabricating e-skin with stretchability, self-healing ability, and

biocompatibility. The unique combination of gluten and EGaIn successfully creates a robust gluten network with improved strength and toughness. Moreover, based on the incorporation of EGaIn, the obtained E-GES has an enhanced self-healing ability, and there are no adverse reactions when E-GES is attached to rabbit skin in animal experiments. E-GES not only well matches the requirements of an ideal e-skin but also demonstrates acute strain-sensing ability in different situations, ranging from large-scale human motions to tiny strain changes. This work provides an attractive idea to design e-skin through the construction of protein networks by liquid metal, clearly demonstrates the self-healing mechanism based on the use of dynamic bonds also suitable for endowing protein-based e-skins with self-healing ability and successfully shows the vast potential of protein-based e-skin in sensing human motion. This method could be further developed by using other proteins, such as gelatin and egg albumin,

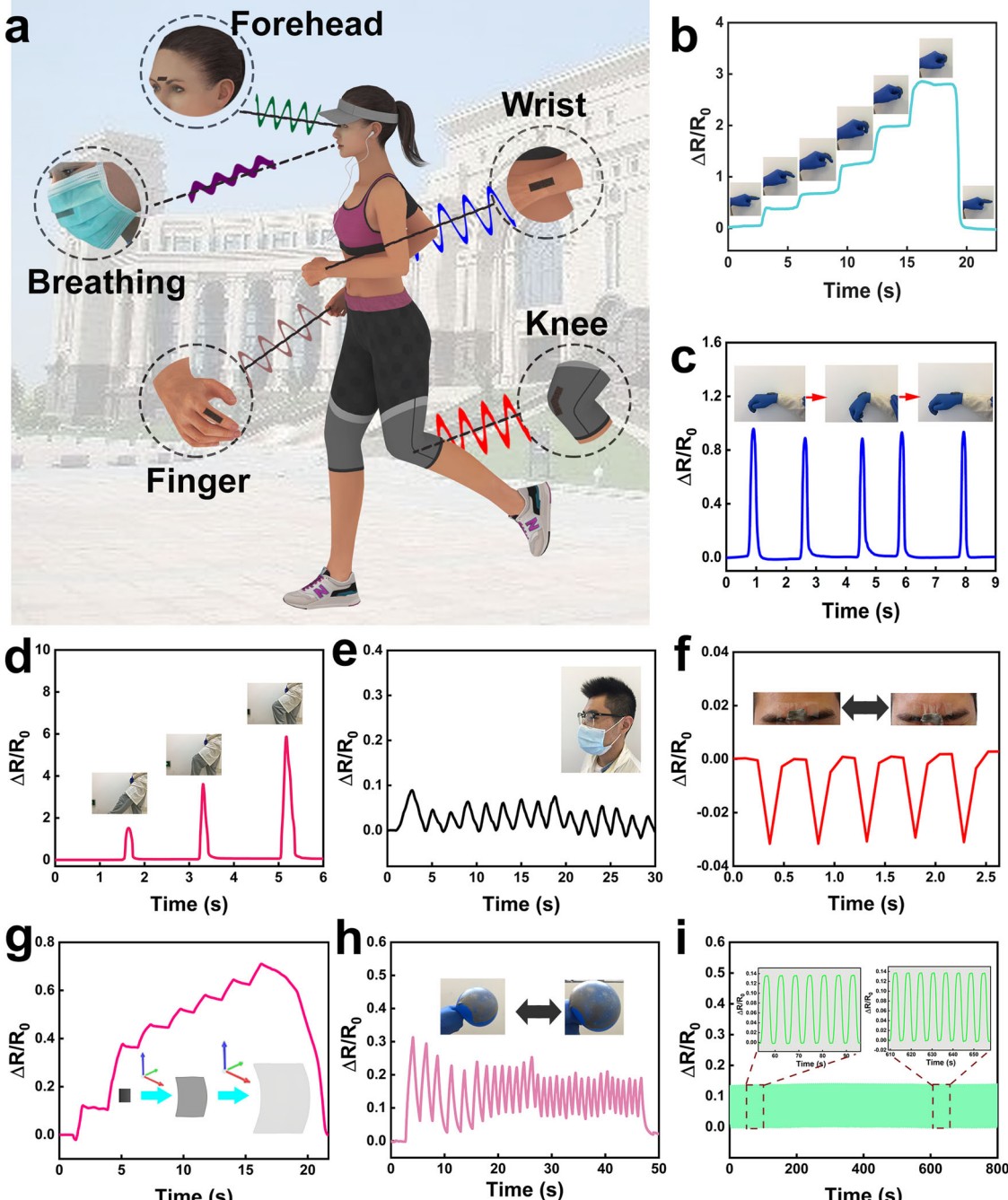

**Fig. 5 Characterization of the strain-sensing ability of E-GES. a** Schematic illustration of strain sensing tests on different body parts. **b, c, d** Real-time resistance changes of E-GES when adhered to the forefinger (**b**), wrist (**c**) and knee (**d**). Inset, photographs of the bending process of the forefinger (**b**), wrist (**c**) and knee (**d**). **e** Real-time resistance changes of E-GES when embedded in a mask. E-GES can detect minor strain changes from breathing. **f**, Real-time resistance changes of E-GES when adhered to the middle of the forehead. E-GES can monitor deformations of facial expressions. **g, h** Real-time resistance changes of E-GES when adhered to the surface of a balloon. Inset, schematic illustration of the increasing strain change of E-GES in three dimensions. (**g**) and photographs of E-GES on the surface of the balloon (**h**). **i** Stability tests of E-GES under stretching at 50% for more than 100 cycles.

and other inorganic materials, such as MXenes, carbon nanotubes, and silver nanofibres, to obtain an e-skin with desired functions.

## Methods

**General methods.** Gluten (pure substance) and EGaIn were purchased from the Meryer (Shanghai) Chemical Technology Co., Ltd. and Shenyang Jiabei Commerce Ltd., respectively. Guanidine hydrochloride, glycine (gly), trichloroacetic acid, β-mercaptoethanol, NaCl, and methyl cellulose M20 (MC) were obtained from Sinopharm Chemical Reagent Co., Ltd. Tris (hydroxymethyl) aminomethane

(Tris), ethylenediaminetetraacetic acid (EDTA), 5,5′-dithiobis-2,2′-nitrobenzoic acid, and urea were purchased from Genview Scientific Inc., Beijing Solarbio Science & Technology Co., Ltd., Shanghai Macklin Biochemical Co., Ltd., and Tianjin Fengchuan Chemical Reagent Co., Ltd., respectively. The artificial sweat was purchased from Dongguan ChuangFeng Automation Technology Co., Ltd. The above reagents were used without further purification unless otherwise stated. The EGaIn-dispersed solutions were prepared by dispersing different masses of EGaIn in a 1% MC aqueous solution, followed by sonication in an ice bath for 70 min (Supplementary Fig. 21) (ultrasonic cell disruptor JYD-1800L with a 6-mmφ probe, Shanghai Zhixin Instrument Co., Ltd.). To obtain a robust gluten network, a small

amount of NaCl (5.66 wt% of the gluten) was first dissolved in the 1% MC aqueous solution before the addition of EGaIn. The E-GES sample was fabricated by mixing equal quality of the EGaIn-dispersed solution and gluten through a hand mixer and kneading (Supplementary Fig. 22). According to the mass ratio of EGaIn to gluten, the four obtained E-GES samples were denoted as 1%, 3%, 5% and 10%, and the control sample was obtained by mixing MC aqueous solution with gluten, which was denoted as 0%. The E-GES micromorphology was observed by FESEM (Zeiss Ultra55) equipped with EDS, and data was collected and analysed by ZEISS smartSEM User Interface v.6.01 and RemCon 32 software. FTIR spectra were recorded by a PROTA-2X$^{TM}$ FT-IR Protein Analyzer, and the obtained data were analysed through GRAMS/AI v.7.00 software. Raman spectroscopy was performed with a Renishaw In Via Qontor confocal Raman microscope that used a 785 nm excitation laser, and the collected Raman spectroscopy data were analysed with WiRE 5.3 software. XPS was performed using a Thermo Fisher ESCALAB 250Xi with Al Kα radiation. DSC was performed on a TA Q2000, and samples were heated from 10 °C to 150 °C at 10 °C/min under nitrogen flow; data was collected by TA Universal Analysis v.4.5 A software and analysed with Origin 2021b software. The free -SH and S-S contents were measured and calculated according to a previous report[13]. Briefly, to measure the free SH content, a E-GES sample (30 mg) was suspended in 1 mL of Tris-gly buffer solution (10.4 g/L of Tris, 6.9 g/L of glycine, 1.2 g/L of EDTA, pH 8.0) containing 1.91 g of guanidine hydrochloride, and then the obtained solution was diluted to 4 mL and denoted as the sample solution. 2.0 mL of guanidine hydrochloride solution (5 mol/L) and 0.05 mL of 5,5′-dithiobis-2,2′-nitrobenzoic acid (DTNB) solution (4 mg/mL) were added into 1.0 mL of the sample solution to react for 30 min in the dark. Then the absorbance of the reaction solution was recorded at 412 nm against 1 mL of reagent solution as blank to calculate the free SH content (UV-VIS spectrophotometer TU-1810, Beijing PERSEE General Instrument). To measure the total SH content, 4.0 mL of guanidine hydrochloride solution (5 mol/L) and 0.05 mL of β-mercaptoethanol were added into 1.0 mL of the above sample solution to react for 1 h at 25 °C, and then 10 mL of trichloroacetic acid solution (12%) was added to react for another 1 h. The reaction solution was centrifuged (5000 r/min, 10 min) to collect precipitate, and then the collected precipitate was twice resuspended in 5 mL of tri-chloroacetic acid solution (12%) and centrifuged (5000 r/min, 10 min). The purified precipitate was dissolved with 10 mL of urea solution (8 mol/L) before adding 0.1 mL of DTNB solution (4 mg/mL) to react for 30 min in the dark, and then 5 mL of Tris-gly buffer solution was added to 1 mL of this solution to record the absorbance at 412 nm against 1 mL of reagent solution as blank to calculate the total SH content. Free SH and total SH content were calculated according to the following formulas: $SH = (73.53 \times A_{412} \times D)/C$, where $A_{412}$ is the absorbance at 412 nm, $D$ is the dilution factor (3.05 for $SH_{free}$, 10.01 for $SH_{total}$), and $C$ is the sample concentration in mg/mL. S-S bonds content was calculated according to the following formulas: S-S bonds = $(SH_{total} - SH_{free})/2$. Data were analysed with Microsoft Excel 2019 and Origin 2021b. The E-GES samples for the above tests were lyophilized before testing, whereas the following measurements directly used E-GES samples.

**Mechanical tests**. For mechanical tests, the E-GES samples were shaped by a custom-made cylindrical PTFE mould with an inner diameter of 15 mm and a height of 15 mm according to the method of H.C.D. Tuhumury et al[27,28]. with some modifications. After carefully removing them from the mould, the shaped samples were glued onto pneumatic grips and tested with an Instron 5966 tensile tester equipped with a 10 N load cell at a raising rate of 300 mm/min. Data was collected by Instron bluehill universal v.4.08 software and analysed with Origin 2021b software. The mechanical properties of E-GES samples (i.e., the maximum tensile stress, tensile toughness and Young's modulus) were analysed through the obtained stress-strain curves, and the recovery rates were calculated according to the following formulas: stiffness recovery=$E_2/E_1 \times 100\%$, toughness recovery=$A_2/A_1 \times 100\%$ ($E_1$ and $A_1$ were the Young's modulus and dissipated energy of the original E-GES, while $E_2$ and $A_2$ were the Young's modulus and dissipated energy of the self-healed E-GES).

**Rheological tests**. Rheological measurements were performed on an Anton Paar Physica MCR 301 rheometer with a 25 mm parallel-plate geometry and a fixed gap of 1 mm at 25 °C. The linear viscoelastic region was obtained by oscillation measurement with strain values changing from 0.1 to 1000% at a constant frequency of 1 Hz, and then, frequency sweeps were carried out in a frequency range of 0.1 Hz to 100 Hz at a fixed strain of 0.5%. Data collection and analysis were carried out with Rheoplus software v.3.21 software.

**Biocompatibility tests**. A leaching pattern test of human epidermal keratinocyte cell lines (HACAT cells) and human dermal fibroblast cell lines (HSF cells) for E-GES was used to evaluate the cytocompatibility of E-GES based on a previous reference[52]. The cytotoxicity of E-GES was measured by employing a leaching pattern test of HACAT cells and HSF cells for E-GES. First, E-GES (approximately 4cm$^3$) was lyophilized, and sterilized by UV for 5 h to obtain dried E-GES (dE-GES). Sterilized dE-GES extract solutions with dE-GES concentrations of 5 mg/mL and 10 mg/mL in Dulbecco's modified Eagle's medium (DMEM, GIBCO) were prepared by immersing dE-GES in the medium for 24 h in an incubator at 37 °C in

5% CO$_2$. Meanwhile, HACAT cells and HSF cells were precultured for 12 h before replacement with a series of different groups of dE-GES extract solutions. The cell proliferation, viability, and apoptosis were tested by Alamar Blue (AB) assay, a LIVE/DEAD Viability kit, and an AnnexinV-FITC/PI Apoptosis detection kit. For cell proliferation experiment, HACAT cells and HSF cells were cultured in 96-well plates at a density of 5000 cells/well with 5 mg/mL and 10 mg/mL dE-GES extract solutions. After being cultured for 24 h, 1/10$^{th}$ volume of cell viability reagent was added directly to cells before being incubated for 4 h. Finally, the absorbance of cells was recorded at 570 nm against 600 nm as a reference wavelength in Multiskan FC microplate reader (Thermo scientific, China). For cell viability experiment, HACAT cells and HSF cells were cultured in a confocal dish with 10 mg/mL dE-GES extract solutions for 24 h, and live cells and dead cells were stained with Calcein-AM and EthD-I, respectively for 5-25 min. The results of live cells (using excitation wavelength of 490 nm) and dead cells (using excitation wavelength of 528 nm) were recorded by in C2 + confocal laser scanning microscope (Nikon, Japan) with NIS-Elements AR 4.20 software. For the cell apoptosis experiment, HACAT cells and HSF cells were cultured in a culture flask with 10 mg/mL dE-GES extract solutions for 24 h, and then were resuspended in 1X binding buffer at a concentration of 1×10$^6$ cells/mL. After adding 5 μL FITC Annexin V and 5 μL PI, the cells were gently vortexed and incubated for 15 min in the dark. Finally, 500 μL 1X binding buffer was added to the above solution to analyze the cell apoptosis by Gallios flow cytometry (Beckman Coulter, America) and the obtained data were analysed by Gallios cytometer v.1.2 and FlowJo V.10 (TreeStar) software. Gating strategies are shown in Supplementary Fig. 23. In addition, to investigate whether EGaIn can leak from E-GES when E-GES is immersed in the culture solution, inductively coupled plasma (ICP) spectrometry (ICP-OES, Agilent 5110) was used to determine the concentration of the leaked liquid metal in the culture solution containing E-GES. In previous biocompatibility tests, we only placed one piece of E-GES in the culture solution, so to increase the chances of EGaIn being released, we kept the weight of the immersed E-GES unchanged but cut it into smaller pieces (Supplementary Fig. 24). The ICP results show that although the culture solution contains 10 mg/mL E-GES, the concentration of Ga in the solution is 4.9×10$^{-4}$ mg/mL, and the concentration of In is almost undetectable (Supplementary Fig. 25). Considering that the concentrations of Ga and In are extremely low in both culture solutions, leakage of EGaIn from E-GES after suffering damage can be inferred to be almost impossible. For on-skin biocompatibility tests, five dorsal areas (approximately 8 cm$^2$ of each area) of experimental rabbits (four 1-year-old health Chinese white rabbits, 2 male and 2 female, approximately 3 kg of each, $n = 4$) were partially depilated before testing, and the depilated dorsal areas were named according to their positions on the back. The upper left dorsal area, the upper right dorsal area, the lower-left dorsal area and the lower right dorsal area were named as area 1, area 2, area 3, and area 4, respectively. The dorsal area that was directly above area 1 and area 2 was named as area 0, and was used as control without any treatments. #-shaped wounds were introduced into area 2, area 3, and area 4, respectively. The extraction experiment was carried out by immersing E-GES samples in artificial sweat (at an extraction ratio of 0.1 g:1 mL) at 37 °C for 24 h. The E-GES samples were firmly attached to area 4 by using a bandage, and area 3 was used to observe the wound naturally recovering process. Treated sweat was dripped on area 1 and area 2 (Supplementary Fig. 26). These on-skin biocompatibility experiments lasted for 4 h every day and continued for 7 days to evaluate the biocompatibility of E-GES. All animal experiments were performed according to the guidelines for laboratory animals established by Fudan University and conducted according to the Institutional Animal Care and Use Committee (IACUC) guidelines.

**Degradation tests**. For degradation tests, a pepsin solution was prepared by dissolving 0.3 g of pepsin in 30 mL of 0.1 mol/L HCl solution. The degradation tests were carried out by immersing E-GES samples in the obtained pepsin solution and moist soil at room temperature.

**Strain sensing tests**. For strain sensing tests, the real-time R-T curves were recorded by a Keithley 2604B SourceMeter instrument. The E-GES-based strain sensor was prepared by using E-GES as the conductor and commercial VHB tape as the encapsulant. During the strain sensing tests, the tensile strain of the tested samples was controlled by customized motorized long-travel translation stages. Data were collected by Kickstart v.2.5.0 software and analysed with Origin 2021b software.

**Reporting summary**. Further information on research design is available in the Nature Research Reporting Summary linked to this article.

## Data availability
All data in main text and supplementary information is available from the corresponding authors upon reasonable request.

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

## Acknowledgements

We deeply thank Professor Hongdong Zhang for evaluating our work design, Professor Shaoning Yu for helping us analyse gluten secondary structure, Professor Zhanfeng Yang for discussing mechanical experiments with us, and Yuning Wang for her spiritual support for us. This work was financially supported by National Natural Science Foundation of China (51972064).

## Author contributions

B.C. and Y.C. conceived and designed the work, prepared E-GES, conducted sensing experiments and contributed to manuscript writing. Q.L. was responsible for the design and analysis of cytotoxicity tests as well as the manuscript writing of this part. Z.Y., M.W., and Y.Q. performed the rheological measurements, DSC, SEM, analyzed related data and contributed to the explanation of mechanical performances of E-GES. R.L. and Y.Z. conducted -SH and S-S contents tests, related data analysis and contributed to the interpretation of cross-linking mechanism. Z.C., W.Y., C.S., and Z.L. conducted the Raman spectroscopy, FTIR spectroscopy, analyzed related data, interpreted the secondary structure reconfiguration of gluten and were responsible for all schematic illustration drawings of E-GES network structure. M.Y. and J.S. performed tensile experiments and funded this work. P.D., M.O.L.C., X.Z., and P.M.A. edited the manuscript, took part in the discussion of results and provided many professional suggestions about improving the experimental design of E-GES.

## Competing interests

The authors declare no competing interests.
