## [Peer Review File · Nature Communications]

Liquid metal-tailored gluten network for protein-based e-skinREVIEWER COMMENTS

Reviewer #1 (Remarks to the Author):

Recommendation:

I do not recommend this manuscript for publication in Nature Communications. It can be considered in other specific journals.

The presented work entitled "Liquid metal-tailored gluten network for protein-based e-skin" submitted by Chen et al. reports a highly stretchable gluten network based conductive material and its application towards self-healable electronic skin. The combination of gluten and E-GaIn plays a key role in creating robust protein networks with high toughness and stretchability. In order to show the perspective of the application, the authors also demonstrated biocompatible and biodegradable e-skin capable of sensing strain, ranging from large-scale human motions to tiny strain changes. Since this manuscript, however, lacks some experimental interpretation and clarity on sensing mechanism, I would like to point out that the authors should modify this manuscript for the possible readers to better and more richly understand their work.

1. The authors have repeatedly demonstrated that the sample has excellent elastic recovery ability, not only having good tensile properties, but also giving E-GES a stable electrical signal change after recovery from deformation. For example, the clear and repeatable resistance signal has been displayed by E-GES under the deformation of the fingers and wrist (Figure 5c), the stable electrical signal cycle test (Figure 5i), and by the strain sensitivity to slight movements such as blinking (Figure 5f). However, the stable recovery ability of the above E-GES is demonstrated only by the electrical signal test. In the mechanical tensile test, the authors dealt with a single-cycle tensile test only. I recommend that the sample should be subjected to more cycles of tensile testing under the most commonly used strain, in order to match the electrical signal test results.
2. The authors have mentioned that E-GES as an electronic skin is inevitably exposed to sweat. That being the case, it is necessary to explore the self-repair ability of sweat/environmental humidity. At the same time, it is also necessary for authors to further investigate the stability of E-GES materials and its electrical properties under sweat/environmental humidity.
3. The authors showed the recovery of mechanical and conductive properties, differing under various conditioning in temperature in the Fig. 3e and Supplementary Fig. 9. However, recovery mechanism and hypothesis depending on temperature are not fully verified. Moreover, it is so hard to find out both of quantitative analysis of conductive properties and time range of low temperature conditions. Additional experiments are required to support this study.
4. The biocompatibility test should be performed on the leaching solution of the damaged E-GES. Because when it is attached to a living body as an electronic skin, if the E-GES suffers a deep incision, liquid metal will leak. Although it has shown that liquid metal has good biocompatibility, it is necessary to cut the E-GES in the culture solution and wait for it to heal before testing the biocompatibility of the leaching solution.
5. In the manuscript, the authors mentioned that "human skin generally suffers from around 30% of the tensile strain in daily human motions, and some reports suggest that the skin-like wearable flexible hybrid electronics with stretchability as high as 100% are in favor of obtaining high-quality signals from skin", showing that the stretchability of E-GES is comparable to that of synthetic materials in figure 2c. However, as the authors also noted, flexible hybrid electronics with stretchability as high as 100% are sufficient to be used as the electronic skin for detecting human motions. From this point of view, several protein-based materials in figure 2c also satisfy the tensile strain to detect human motions. Therefore, the authors should mention the differences with these established protein-based materials and clearly explain the need with more persuasive allegations for high stretchability.
6. In this paper, the authors described a material design that could address the limitations of engineering protein structures to realize self-healing electronic skin. However, the interpretation of the self-healing characteristics is unclear. The authors need to provide more additional descriptions and results to clarify the following contents list below:
 - i) In Figure 3c, the authors described three self-healing mechanisms of E-GES network. But, the interpretation of the contribution of each mechanism is uncertain. Therefore, the authors should

present a clear interpretation by adding self-healing properties when only each mechanism exists.
ii) The authors noted the effect of the tyrosine side-chain of gluten for self-healing characteristics. However, the interpretation of the self-healing mechanism via tyrosine in E-GES is lacking. Therefore, the authors should provide the chemical structure with additional NMR data for the tyrosine group in E-GES. The authors should then supplement the additional schematics and explanations of the self-healing mechanism.
7. Please check for some typo and figure errors. (Figure S7)

Reviewer #2 (Remarks to the Author):

Dear Authors

Overall, this work is an interesting study that possibly meets with the interest of the scientifically broad readership of Nature Communications. The result shows that relatively small amounts of Gallium and Indium can be used to actively control the crosslink density of the protein molecules due to the ability of the metals to interact and occupy the -SH thiol groups, creating metal ligands while reducing crosslink density and providing a softer material with stretchability. The story is relatively straightforward, and the novelty is high. No similar studies have been found in the literature.

The topic in itself seems timely selected in a more dominant science related to biopolymers and their investigated use as alternative resources on the market with more sceptics towards petroleum-based engineering plastics. It is believed that the set of experiments provided comprises in a relatively short format a good story to the readership, while at the same time displaying potentials with natural materials as such. It is in this sense inspiring reading that will probably receive follow-up work and citations.

Occasionally, it is difficult to distinguish between speculative statements and statements that the authors have based on their findings. This may be a consequence of technical proofreading carried out by someone that only have corrected the language but has not comprehended the message in what was said in the text. I have marked out these sections so that the authors can address this problem, but it is still recommended that Nature Communications proofreading services review the text.

The list of comments provided, line by line, was made because the manuscript can be considered for publication in a revised format. The recommendation from my side, therefore, is a major revision is required.

Finally, the number of authors is 18! It is impossible to understand how it could have been necessary with several scientists to carry out simple IR verifications. The reviewer knows that the first three authors contributed equally, and the last two supervised the work. The rest of the authors need detailed motivation for their contribution since the total number of authors is almost twice as many as usual for these works in Nature Communications.

Yours sincerely,

List of comments line by line:

1. Abstract:

Line 21 and 31: Biocompatibility? Allergenic people? What is meant?

Line 27: Hierarchical Sulphur bonds... -What do the authors mean by hierarchical? If the authors mean that the disulfide bonds are subordinate in affecting mechanical properties (compared to the beta-sheets), this needs to be clarified.

Line 30: To avoid confusion: "a synthetic natural like stretchability" is mentioned, but it confuses the reader to believe that stretchability refers to a synthetic (non-biobased) polymer material, instead of highlighting the stretchability integrated into the material as a property displaying stretch/ recovery values similar to, e.g., natural rubber. Also, see articles on natural rubber as sensors.

2. Introduction:

Line 40: outstanding mechanical properties deserve to be defined, e.g., as compared to what?

Line 47: no need to refer to microstructure. Only structure is sufficient.

Line 52-54: protein is a polymer in itself. Simplify text to be more accurate and to the point.

Line 57: 'loses,' is it meant 'misses'?; check with proofreading company.

Line 60: Gluten is typically a hard/rigid network when no plasticizer is included in the consolidated material (e.g. H₂O), the statement about strengthening and toughening is awkward in the context of using EGaIn to break up the network. Only toughening likely applies here, needs better explanation otherwise, or reference, missing reference?

Line 65-75: rephrase the entire section for clarity. The "reversible -SH/S-S exchange mechanism" needs referencing and /or clarification. The -SH/S-S links are known to be possible to rearrange.

Line 72: s-s bonds maintain the structural integrity while EGaIn -sh coordination bonds contribute to dissipating energy. This should be referenced or clear to the reader that it is a speculative proposed theory of the authors.

Line 75: The authors mean an increase in Beta -sheet contents

Line 88-97: The referencing should appropriately highlight the previous research on gluten materials, referencing representative research over the last 30 years instead of referencing a few works since 2014, since this knowledge was known long before.

Line 85: "for developing"

Line 97: How do the authors confirm no starch? Check materials section. No purification steps were described. At least the protein contents should be provided in the materials section.

Line 100: small mass? NaCl? 1 wt.%, 3 wt.% etc

Line 114: The text in Fig 1b and c are too small to see properly. Central information in Fig 1 should be enlarged. Non-important popular scientific imaging can be made smaller.

Line 115: 'support these observations' – better English

Line 116: It is unclear what Fig 1d-g refers to, the legend of the entire Fig 1 needs revision for improved clarity.

Line 127-129: redundant - remove

Line 130-132: the values should be given some comparison, for example: How do these values compare to commercial polymer for skin coverage such as PDMS patches of silicone?

Line 135-136: ...minimum value of 1200%, see fig 2a for 10 wt.%EGaIN.?

Line 145-146: ...alfa-helix in all EGES samples...: here is needed clarification because, at this

point, the reader has not been introduced to alfa-helix molecular conformations. The % of alfa-helix molecular conformations is mentioned and compared for previous works, but no value of the % is provided for the E-GES. Why? High content? ...or is it only that poor English was used and the content was determined to 20-30%? As it stands now, these values are related to previous studies.

Line 148-155: In this section, the authors need to introduce the reader to the difference in conformation between alfa helices and the beta sheets and provide relevant references on these matters. Not every reader is familiar with these conformational differences, and it is also necessary since the authors in the next section present thermal stability results (line 157-159). Alfa helices and beta sheets are only one part of the entire protein macromolecular complex. What about the tertiary and quaternary protein structures?

Line 157-159: "The enhancing thermal stability..." : Here, the authors need to correct their statement to proper English (although the reviewer can see what the authors aim at stating from a scientific results perspective).

Line 160-162: At this point in the text, the authors again refer to polymer physics and how thermal stability can be influenced for specific structures due to more rigid arrangements / stricter molecular conformations, highlighting the need for brief info to the reader about the structures (above). Additional and better referencing should be provided.

Line 165-166: rephrase: "... crosslinking sites synergistically promote the adjustment of the EGes network structure". The meaning of this is unclear.

Line 169-174: proper referencing to recent and most informative articles on how rheological behaviour vary for semisolid protein hydrogels in relation to the presence of metal ions is required. The authors currently only relate mechanical properties change to crosslinks and beta sheets, whereas in reality, 10% metal ions have been introduced. See reference: doi.org/10.1021/acsnano.0c10893

Line 171: English problem: The improving... It should state: The enhanced... The use of a proofreading agency that actually reads what is being said in the article is necessary. It is clear that the work has been through professional proofreading but not with a language critical person who has understood the meaning of the sentences (sometimes).

Line 177: remove the word *microscopic*. It has no meaning here.

Line 188-190: This statement needs to be proven and shown with a photo.

Line 203-206: The section needs clarification because it is not possible to understand the specifics. What is a circular wound?

Line 210: remove poor English, 'is deeply', write: was

Line 214: what is 'ratio valve'?

Line 217-226: the authors need to be clear about that they speculate on the mechanism and that their suggested mechanism is based on the xx and yy observations. Are the authors talking about gluten protein denaturation in the presence of EGaIn? Have the authors reflected over which one of the gluten protein fractions (e.g. gliadin or glutenin) that dominantly is exposed in the self-healing mechanism? Gliadin and Glutenin have very different amino acid compositions.

Line 228: in a great measure = to a great extent (English)

Line 231: status should read experiments

Line 234: indicating the electrical stability as a consequence of elasticity of the polymer network, it

should read.

Line 236: again, the English need revision from someone skilled in technical writing, e.g. 'after suffering from complete cutting'

Line 243 to 285 requires a reviewer knowledgeable in the fields of biocompatibility. From this reviewer's perspective, it is unclear how these materials are used with people showing allergic reactions towards gluten protein. 6-7% of the earth population has been estimated to suffer from gluten sensitivity.

Line 286-309:

What are authors reflections about similar works on strain sensors such as this work,

<https://www.nature.com/articles/ncomms4132>

and how can the authors relate their work to previous works on strain sensors? Overall, the referencing in the manuscript is rather shallow.

Line 311: Have the authors created a new protein?

The point-to-point responses to reviewers' comments

Reviewer #1 (Remarks to the Author):

Recommendation:

I do not recommend this manuscript for publication in Nature Communications. It can be considered in other specific journals.

The presented work entitled “Liquid metal-tailored gluten network for protein-based e-skin” submitted by Chen et al. reports a highly stretchable gluten network based conductive material and its application towards self-healable electronic skin. The combination of gluten and EGaIn plays a key role in creating robust protein networks with high toughness and stretchability. In order to show the perspective of the application, the authors also demonstrated biocompatible and biodegradable e-skin capable of sensing strain, ranging from large-scale human motions to tiny strain changes. Since this manuscript, however, lacks some experimental interpretation and clarity on sensing mechanism, I would like to point out that the authors should modify this manuscript for the possible readers to better and more richly understand their work.

A: Thanks for your forward-looking and constructive suggestions for improving our job.

(1) **First of all, we would like to introduce the design philosophy and main concern of this work.** Since General Electric built the first sensitive skin for a robotic arm in 1985, significant progress in endowing e-skin with human skin-like mechanically compliant yet highly sensitive properties has been achieved in recent

years¹. However, although proteins are the vital components of skin and ensure the multiple capabilities of skin, and thus, designing protein-based e-skin might be one of the most powerful forces to further drive the development of e-skin, the research on protein-based e-skin is still in its infant stage. Reasons for this phenomenon may be that the composition and structure of proteins are far more complicated than that of polymers with repetitive structure units, which means regulating and controlling the structure of protein to meet the requirements of e-skin preparation is a tough job. After all, nature has taken countless years of evolution to create the skin, the largest sensory organ in our bodies. The dominant materials used for preparing e-skin belong to petrochemical materials, such as poly (vinyl alcohol) (PVA)², poly (ethylene glycol) (PEG)³, polydimethylsiloxane (PDMS)⁴, polyurethane (PU)⁵, etc. Currently, the most common protein used for e-skin is silk fibroin (SF), but due to the huge difference in the mechanical properties between SF and the skin, some complex preprocessing is required to adjust the structure of SF and the reported SF-based e-skins often lack the self-healing ability⁶⁻⁸. **Hence, we hope to find another alternative that possesses better performance after simpler treatments to replace SF for making e-skin.** Moreover, from the standpoint of environmental protection, proteins are more easily degraded than most petrochemical materials in nature, thus benefiting for reducing the electronic waste pollution and the dependence on fossil energy sources. Based on the reasons mentioned above, we spent the last three years looking for appropriate protein as raw material and carrying out various explorations to design a new protein-based e-skin, and **we believe that this work is meaningful and valuable.**

Based on our research results in recent years, we reported a gluten-based e-skin, namely E-GES, in this manuscript and exhaustively discussed the design strategy that we used to improve the gluten structure through analyzing the chemical environment of EGaIn in the gluten network and the change of gluten backbone conformation. Since an attractive material for e-skin preparation should process stretchability, self-healing ability and biocompatibility as Bao et al. reported in their review paper about e-skin⁹, we think that **to prepare protein-based e-skins, the first and foremost essential step is to tailor the protein structure to meet these basic requirements** and thus we especially focused on the explanation of the protein structure regulation mechanism in this work. However, we deeply know that there are still a lot of difficulties for us to overcome to develop more protein-based e-skins in the future and one of these difficulties is to investigate the sensing mechanism of different proteins. Therefore, we think that your opinions are very to the point and we really appreciate your professional advice for improving our work. Thank you very much for your valuable guidance and great kindness.

(2) **As for the sensing mechanism for using E-GES as strain sensor, E-GES actually can be classified as a kind of ionic conductive hydrogels**, namely that the polymeric network structure of E-GES makes it solid-like, while the aqueous phase of E-GES enables the diffusion of ions. The physical deformation of E-GES caused by strain will lead to the change of the network structure and thus the variation of the resistance of ion migration¹⁰. Besides, this mechanism has been reported on the research for dough-based e-skin. In this report, Lei et al. pointed out that electrolyte salts (e.g.,

NaCl and K_2CO_3) in dough can serve as ion carriers, so dough can be used to fabricate e-skin to detect human motions¹¹. It is well accepted that the gluten network is the main structural framework of dough¹², so the migration of ions actually occurs in gluten network. Therefore, it is clear that **the sensing mechanism of gluten-based e-skin is consistent with that of dough-based e-skin.**

We are sorry for the lack of relevant explanations for the sensing mechanism of E-GES in previous manuscript, and we have supplemented this part of content and the corresponding schematic illustration in the revised manuscript for you to reevaluate our job. The schematic illustration of the sensing mechanism of E-GES is proposed and shown in Fig.R1. When E-GES is stretched under applied stress, the deformation also occurs in its internal network structure, so the migration path of ions changes accordingly and thus resistance values vary. On the contrary, when the network structure of E-GES recovers after releasing the stress, the resistance values decrease. We hope this improvement will enable readers to understand the sensing mechanism of E-GES. We sincerely apologize for this important problem, and we deeply appreciate your professional opinions. Thank you very much.

Besides, it should be noted that the addition of NaCl is necessary for E-GES preparation. Since the gluten protein has net positive charge due to the higher distribution of the basic amino acid residues, including arginine, histidine and lysine¹³, the positive charge at the protein surface makes the protein molecules repel each other and thereby reduces protein-protein interaction resulting in a weaker network structure^{14,15}. However, when salt is present, it shields the charge on the gluten protein,

thereby reducing the electrostatic repulsion between protein molecules and thus, allowing them to associate more closely and produce a stronger protein network^{16, 17}. The studies about the interaction between salts and gluten are well-developed nowadays, and we learn from the results of these studies to design E-GES.

Fig. R1 | Schematic illustration of the sensing mechanism of E-GES.

(3) At last, we think the emphasis of this paper is to present a reliable strategy to tailor the protein structure by using liquid metal and to explain the interaction between protein and liquid metal through a series of characterization results, **because achieving the design of protein structure is the basic step for preparing protein-based e-skin.**

We believe that this work can provides some enlightening thoughts and concepts for developing more different proteins to design protein-based e-skins, such as gelatin and egg albumin. Besides, since “Nature Communications” is a diverse and inclusive platform with a large number of readers from different academic backgrounds and it also is a place of creativity and inspiration, we hope that our manuscript could be

published on “Nature Communications” to share our ideas with other scientists and if possible, we look forward to receiving different opinions about our work. Therefore, **we greatly appreciate your constructive suggestions from a standpoint of improving the design of our work.**

We admit that the previous version of our manuscript does have many problems such as the imperfect logical structure in some sections, some wrong English expressions and so on, so we fully understand your decision about our manuscript. However, after correcting these problems, we have also tried our best to improve our manuscript according to your opinions and we have double checked the contents in all sections. Therefore, we believe that the revised version of the manuscript can meet the publication requirements of “Nature Communications”. We are grateful to your reconsideration of our manuscript, and we look forward to receiving any comments from you. We are honored to discuss with you and we have learned a lot from your comments. Thank you very much.

Supplementary content:

The electrolyte salt NaCl was added to E-GES during the preparation process, so it can serve as an ion carrier, thus endowing E-GES with the ability to be used as a strain sensor, relying on the resistance changes to detect variations in applied strains (Supplementary Fig. 12). This sensing mechanism has been proven to be effective in the gluten network in dough-based e-skin¹¹.

Supplementary Figure 12. Schematic illustration of the sensing mechanism of E-GES. E-GES actually can be classified as a kind of ionic conductive hydrogels, namely that the polymeric network structure of E-GES makes it solid-like, while the aqueous phase of E-GES enables the diffusion of the ions. The physical deformation of E-GES caused by the strain will lead to the change of the network structure and thus the variation of the resistance of the ion migration⁶. When E-GES is stretched under applied stress, the deformation also occurs in its internal network structure, so the migration paths of ions change accordingly and thus the resistance value varies. On the contrary, when the network structure of E-GES recovers after releasing the stress, the resistance value decreases.

Q1. The authors have repeatedly demonstrated that the sample has excellent elastic recovery ability, not only having good tensile properties, but also giving E-GES a stable electrical signal change after recovery from deformation. For example, the clear and

repeatable resistance signal has been displayed by E-GES under the deformation of the fingers and wrist (Figure 5c), the stable electrical signal cycle test (Figure 5i), and by the strain sensitivity to slight movements such as blinking (Figure 5f). However, the stable recovery ability of the above E-GES is demonstrated only by the electrical signal test. In the mechanical tensile test, the authors dealt with a single-cycle tensile test only. I recommend that the sample should be subjected to more cycles of tensile testing under the most commonly used strain, in order to match the electrical signal test results.

A1: Thanks for your comments. **We think this experiment is important for discussing the stability of E-GES in sensing ability. We deeply appreciate your professional suggestions.** Thank you very much.

We have supplemented cyclic tensile loading-unloading tests of E-GES under 50% strain to support the electrical signal cycle test results in the manuscript. Fig. R2 shows E-GES was subjected to 20 consecutive stretching cycles without any resting time. An obvious hysteresis was observed at the first cycle, indicating the destruction of E-GES gluten network, which is a normal phenomenon due to the lack of time for the broken network to recover¹⁸⁻²¹. The loop areas decrease for the second cycle, and change slightly for the following cycles. The residual stresses and strains remain almost unchanged at last, suggesting E-GES possesses good recoverability.

Fig. R2 | The successive tensile loading-unloading tests of 5% E-GES under 50% strain.

Supplementary content:

Finally, cyclic stretching-releasing sensing experiments demonstrate the stable and recoverable sensing ability of E-GES (Fig. 5i), which results from the good recoverability of the E-GES network (Supplementary Fig. 20). Hence, E-GES is robust and has the ability to resist mechanical damage in daily use, comparable to the robustness of e-skins prepared from synthetic materials, such as PDMS and poly(acrylic acid)^{35,51}.

Supplementary Figure 20. The successive tensile loading-unloading tests of 5% E-GES sensor under 50% strain at a raising rate of 100 mm/min. 5% E-GES sensor was subjected to 20 consecutive stretching cycles without any resting time. An obvious hysteresis was observed at the first cycle, indicating the destruction of E-GES gluten network, which is a normal phenomenon due to the lack of time for the broken network to recover¹⁸⁻²¹. The loop areas decrease for the second cycle, and change slightly for the following cycles. The residual stresses and strains remain almost unchanged at last, suggesting E-GES possesses good recoverability. Therefore, the recoverability of E-GES network guarantees the stable electrical signals detection ability.

Q2. The authors have mentioned that E-GES as an electronic skin is inevitably exposed to sweat. That being the case, it is necessary to explore the self-repair ability of sweat/environmental humidity. At the same time, it is also necessary for authors to further investigate the stability of E-GES materials and its electrical properties under sweat/environmental humidity.

A2: Thanks for your comments about investigating the performances of E-GES under the impact of sweat.

We have supplemented relevant experiments and discussions in manuscript (Fig. R3). In the self-healing test, the cut E-GES was daubed with artificial sweat including the cut interface to explore its self-healing ability. The stress-strain curves show that under the impact of artificial sweat the healed E-GES recovers 70.9% of its stiffness and 39.5% of its toughness after healing from complete cutting (Fig. R3a). Compared with the results of self-healing tests without the introduction of artificial sweat, it is clear that the self-healing ability of E-GES decreases. The reason for that is because the artificial sweat contains different salts, such as Na_2SO_4 , Na_3PO_4 , and Na_2HPO_4 , and according to the studies about the effects of the Hofmeister salt series on gluten network formation, different cations and anions have different influences on gluten network formation^{14, 15}, so the existence of different salts in the artificial sweat will influence the reconstruction of E-GES network and causes a decrease in its self-healing ability. Exploring the influence of the combination of different salts to the gluten network formation is beyond the research idea of this work, so we are sorry that we cannot provide the detailed self-healing mechanism of E-GES in artificial sweat.

As for the electrical properties of E-GES under sweat, Fig. R3b shows that the resistance of E-GES increases during stretching and then decreases during releasing and no obvious conductive hysteresis is observed in this process. In the repeated stretching-releasing cycles, the signal changes of E-GES also keep stable (Fig. R3c). Therefore, it can be inferred that when E-GES is exposed to sweat, its strain sensing

ability will not be affected. Moreover, as shown in Fig. R3d, even if E-GES is mostly immersed in the artificial sweat, the LED bulb keeps lighting up, suggesting the good electric conductivity of E-GES in artificial sweat. We deeply appreciate your opinions about designing relevant experiments to investigate the performances of E-GES under sweat, and we think these ideas are brilliant. We are sorry that we neglected relevant research in this aspect and we believe that your suggestions make our work more complete. Thank you very much.

Fig. R3 | The performances of E-GES under the impact of artificial sweat. a. Stress-strain curves of pristine (black) and self-healed (red) 5% E-GES samples. The cut E-GES was daubed with artificial sweat including the cut interfaces to investigate the self-healing ability. **b.** The stretching-releasing cycle with the strain of 100%. **c.** The successive stretching-releasing cycles of E-GES. The E-GES was daubed with artificial sweat on the surface to investigate its strain sensing ability. **d.** Photographs of E-GES connected with a LED bulb. The artificial sweat was added dropwise to the petri dish.

During this process, every drop of sweat dropped on the upper surface of E-GES until nearly the whole of E-GES was immersed in the artificial sweat.

Supplementary content:

In addition, e-skins are inevitably exposed to sweat in daily use, so the performances of E-GES under the impact of artificial sweat was investigated. In the self-healing test, the cut E-GES was daubed with artificial sweat, including the cut interface, to explore its self-healing ability. The stress-strain curves show that under the impact of artificial sweat, the healed E-GES recovers 70.9% of its stiffness and 39.5% of its toughness after healing from complete cutting (Supplementary Fig. 14a). Compared with the results of the self-healing tests without the introduction of artificial sweat, the self-healing ability of E-GES clearly decreases. The reason for this is that the artificial sweat contains different salts, such as Na_2SO_4 , Na_3PO_4 , and Na_2HPO_4 . According to the studies on the effects of the Hofmeister salt series on gluten network formation, different cations and anions have different influences on gluten network formation^{27,28}, so the existence of different salts in artificial sweat will influence the reconstruction of the E-GES network and cause a decrease in its self-healing ability. E-GES can maintain its strain-sensing abilities well under sweat. Supplementary Fig. 14b shows that the resistance of E-GES increases during stretching and then decreases during release, and no obvious conductivity hysteresis is observed in this process; in the repeated stretching-releasing cycles, the signal changes of E-GES also remain stable (Supplementary Fig. 14c). Therefore, when E-GES is exposed to sweat, its strain-sensing ability can be inferred to be unaffected. Moreover, even if E-GES is mostly

immersed in artificial sweat, the LED bulb keeps lighting up, suggesting good electrical conductivity of E-GES in artificial sweat (Supplementary Fig. 14d). Hence, E-GES has the potential to sense the strain changes from human motions in daily use.

Supplementary Figure 14. The performances of E-GES under the impact of artificial sweat. **a.** Stress-strain curves of pristine (black) and self-healed (red) 5% E-GES samples. The cut E-GES was daubed with artificial sweat including the cut interfaces to investigate the self-healing ability. **b.** The stretching-releasing cycle with the strain of 100%. **c.** The successive stretching-releasing cycles of E-GES. The E-GES was daubed with artificial sweat on the surface to investigate its strain sensing ability. **d.** Photographs of E-GES connected with a LED bulb. The artificial sweat was added dropwise to the petri dish. During this process, every drop of sweat dropped on the upper surface of E-GES until nearly the whole of E-GES was immersed in the artificial sweat.

Q3. The authors showed the recovery of mechanical and conductive properties, differing under various conditioning in temperature in the Fig. 3e and Supplementary Fig. 9. However, recovery mechanism and hypothesis depending on temperature are not fully verified. Moreover, it is so hard to find out both of quantitative analysis of conductive properties and time range of low temperature conditions. Additional experiments are required to support this study.

A3: Thanks for your comments about exploration experiments of E-GES in low temperature conditions. **In fact, when we froze E-GES with liquid nitrogen to shatter it for characterization experiments, we noted that as the temperature of E-GES raises to room temperature, the broken E-GES could be healed when contacting with other broken pieces.** Therefore, we designed relevant experiments to explore whether or not the ultralow temperature could cause permanent damage to the mechanical and conductive properties of the healed E-GES.

Actually, most polymer chains are in their glassy state at such a low temperature (-196.56 °C), including the polymer chains of gluten. Thus, the movements and interactions of these polymer chains are seriously hindered, meaning that to endow polymers with self-healing ability in ultra-low temperature is a challenging work. For E-GES, its self-healing ability disappears when being immersed in liquid nitrogen. However, when its temperature increases, the mobility of gluten chains recovers and then the dynamic chemical bonds can be reconstructed between gluten chains to realize the recovery of gluten network. **Notably, in addition to resisting the mechanical damage, after healing from low temperature damage, E-GES still can restore its**

properties at room temperature. Therefore, we just studied the self-healing ability of E-GES at room temperature, or more precisely, we regarded the liquid nitrogen as a way to destroy E-GES, similar to the knife cutting, and then explored whether or not the healed E-GES could recover its mechanical and electrical abilities at room temperature. **We are so sorry that our bad description about this part makes you misunderstand our research ideas.**

Besides, unfortunately gluten is not an ideal protein for designing e-skin with low temperature self-healing ability, and we think some antifreeze proteins that exist in those fishes living in deep ocean might be the better choice after adjusting their structure. Thus, the research for the self-healing mechanism of E-GES in low temperature is out of the research focus of this work, so we do not focus on the self-healing ability of E-GES in low temperature conditions in this manuscript.

Supplementary content:

Therefore, in addition to mechanical damage, low- temperature damage cannot cause permanent damage to the recoverability of E-GES at room temperature.

Q4. The biocompatibility test should be performed on the leaching solution of the damaged E-GES. Because when it is attached to a living body as an electronic skin, if the E-GES suffers a deep incision, liquid metal will leak. Although it has shown that liquid metal has good biocompatibility, it is necessary to cut the E-GES in the culture solution and wait for it to heal before testing the biocompatibility of the leaching solution.

A4: Thanks for your comments about the biocompatibility of E-GES. Because the E-GES immersed in the culture solution was cut from a whole piece of E-GES, it actually was a damaged E-GES. Therefore, for investigating the liquid metal that leaked in the culture solution, **inductively coupled plasma (ICP) was used to determine the concentration of the leaked liquid metal in the culture solution containing E-GES.** In previous biocompatibility tests, we just put one piece of E-GES in culture solution, so to increase the chances of EGaIn being released, we kept the weight of the immersed E-GES unchanged, but cut it into smaller pieces (Fig. R4). The results show that the culture solutions with different pieces of E-GES possess almost the same concentrations of Ga (Fig. R5). Besides, it should be noted that the maximum concentration of Ga is around 0.49 mg/L, but the minimum detection limit of the utilized ICP instrument is 1mg/L, meaning that the obtained values are inevitably influenced by the instrumental errors, which would significantly affect the accuracy of test results when the obtained values are lower than the minimum detection limit of the instrument. As for In element, it is almost undetectable in ICP test. **It should be noted that the culture solution contains 10 mg/mL of E-GES, but the concentration of Ga in the solution is detected as 4.9×10^{-4} mg/mL.** Considering that the concentrations of Ga and In are extremely low in both culture solutions, **it can be inferred that EGaIn is almost impossible to leak out from E-GES after suffering damage.** Therefore, we believe that the cytotoxicity tests against the human epidermal keratinocytes cell lines and human dermal fibroblasts cell lines are accurate and reliable. Actually, there was the problem of the leakage of EGaIn in the original version of E-GES, but this problem

had been completely solved in the E-GES reported in this manuscript after continual improving the design of E-GES. **We deeply appreciate your professional suggestions and the friendly reminder. We have supplemented these experimental results and relevant explanations for this problem in the revised manuscript. Thank you very much.**

Fig. R4 | The pictures of E-GES immersed in culture solutions. The concentration of E-GES is both 10 mg/mL in two solutions. **a**, The E-GES was cut into two pieces. **b**, The E-GES was cut into four pieces.

Fig. R5 | The concentrations of Ga elements in the culture solutions. It is clear that the concentrations of Ga are lower than the minimum detection limit of the used ICP (1 mg/L). Besides, because when the number of E-GES pieces changes, the concentrations of Ga are almost the same and moreover, the culture solution contains 10 mg/mL of E-

GES, but the concentration of Ga in the solution is 4.9×10^{-4} mg/mL, it can be inferred that EGaIn is almost impossible to leak out from E-GES after suffering damage.

Supplementary content:

In addition, to investigate whether EGaIn can leak from E-GES when E-GES is immersed in the culture solution, inductively coupled plasma (ICP) spectrometry (ICP-OES, Agilent 5110) was used to determine the concentration of the leaked liquid metal in the culture solution containing E-GES. In previous biocompatibility tests, we only placed one piece of E-GES in the culture solution, so to increase the chances of EGaIn being released, we kept the weight of the immersed E-GES unchanged but cut it into smaller pieces (Supplementary Fig. 23). The ICP results show that although the culture solution contains 10 mg/mL E-GES, the concentration of Ga in the solution is 4.9×10^{-4} mg/mL, and the concentration of In is almost undetectable (Supplementary Fig. 24). Considering that the concentrations of Ga and In are extremely low in both culture solutions, leakage of EGaIn from E-GES after suffering damage can be inferred to be almost impossible.

Supplementary Figure 23. The pictures of E-GES immersed in culture solutions.

The concentration of E-GES is both 10 mg/mL in two solutions. **a**, The E-GES was cut into two pieces. **b**, The E-GES was cut into four pieces.

Supplementary Figure 24. The concentrations of Ga elements in the culture solutions. The results show that the culture solutions with different pieces of E-GES possess almost the same concentrations of Ga. Besides, it should be noted that the maximum concentration of Ga is around 0.49 mg/L, but the minimum detection limit of ICP is 1mg/L, meaning that the obtained values are inevitably influenced by the instrumental errors, which would significantly affect the accuracy of test results when the obtained values are lower than the minimum detection limit of the instrument. As for In element, it is almost undetectable in the ICP test. It is clear that the concentrations of Ga are lower than the minimum detection limit of the used ICP (1 mg/L). Therefore, because when the number of E-GES pieces changes, the concentrations of Ga are almost the same and moreover, the culture solution contains 10 mg/mL of E-GES, but the concentration of Ga in the solution is 4.9×10^{-4} mg/mL, it can be inferred that EGaIn is almost impossible to leak out from E-GES after suffering damage.

Q5. In the manuscript, the authors mentioned that “human skin generally suffers from around 30% of the tensile strain in daily human motions, and some reports suggest that

the skin-like wearable flexible hybrid electronics with stretchability as high as 100% are in favor of obtaining high-quality signals from skin”, showing that the stretchability of E-GES is comparable to that of synthetic materials in figure 2c. However, as the authors also noted, flexible hybrid electronics with stretchability as high as 100% are sufficient to be used as the electronic skin for detecting human motions. From this point of view, several protein-based materials in figure 2c also satisfy the tensile strain to detect human motions. Therefore, the authors should mention the differences with these established protein-based materials and clearly explain the need with more persuasive allegations for high stretchability.

A5: Thanks for your comments about the comparisons between E-GES and different e-skins. As the design philosophy of this work that we mentioned above, almost all established protein-based e-skins are designed based on silk fibroin (SF) at present. However, due to the mismatch of mechanical properties between SF and the skin, many complicated strategies, such as plasticization or carbonization, are usually involved to modify the mechanical softness and stretchability of SF for preparing SF-based e-skin and the obtained SF-based e-skins often lack the self-healing ability^{6, 7}. Besides, it should be noted that preparing SF solutions from *Bombyx mori* cocoon fibers is fairly complicated and the obtained SF solutions easily turn bad and thus are not suitable for long-term storage²².

Considering that there are overwhelming quantity and variety of proteins with numerous functions existing in nature, we believe that it should be some fascinating proteins suitable for fabricating e-skin after facile treatments. Therefore, we tried to

design a new protein-based e-skin based on gluten and used liquid metal to improve mechanical properties of gluten network to make it a qualified candidate for preparing e-skin. We find that the stretchability of the obtained E-GES stands out from that of the established SF-based e-skins and is comparable to that of the skin fabricated by synthetic materials. It is sure that the established protein-based materials generally satisfy the tensile strain to detect human motions, **but in the case of robotics, the e-skin with high stretchability will enable the robot to take on a variety of shapes and enable high degrees of freedom in movement**⁹. Therefore, compared with the currently established protein-based e-skins, **E-GES is not only suitable for the detection of human motions, but also has the potential to be used in the robotics**, meaning that with reasonable structural adjustments protein polymers also can be seemed as attractive raw materials for e-skin fabrication.

We are sorry for the lack of comprehensive description about the differences between E-GES and the established SF-based e-skins as well as the meanings of the high stretchability of E-GES. We have supplemented clear explanations about this part for readers to better understand the meaning of our work. We deeply appreciate your professional opinions for making the design ideas of our work clearer.

Supplementary content:

Encouragingly, compared with the low stretchability of currently established protein-based e-skin, the high stretchability of E-GES not only makes it suitable for on-skin sensing applications but also endows it with potential for usage in robotics, as a highly

stretchable e-skin will enable a robot to assume various shapes and achieve high degrees of freedom in movement¹. Importantly, the preparation of gluten-based e-skin is easier than that of the commonly established SF-based e-skin, which can be reflected in the following aspects: (i) extracting gluten from wheat is easier than preparing SF solutions from *Bombyx mori* cocoon fibres³⁶; (ii) the obtained gluten is more stable and easier to store than SF solutions; and (iii) due to the mismatch of mechanical properties between SF and the skin, many complicated strategies, such as plasticization or carbonization, are required to adjust the mechanical properties of SF to prepare SF-based e-skins^{5,7}, whereas the strategy for the design of gluten-based e-skin is far simpler and more convenient. Therefore, gluten can be used as a new attractive protein material for e-skin preparation.

Q6. In this paper, the authors described a material design that could address the limitations of engineering protein structures to realize self-healing electronic skin. However, the interpretation of the self-healing characteristics is unclear. The authors need to provide more additional descriptions and results to clarify the following contents list below:

i) In Figure 3c, the authors described three self-healing mechanisms of E-GES network. But, the interpretation of the contribution of each mechanism is uncertain. Therefore, the authors should present a clear interpretation by adding self-healing properties when only each mechanism exists.

A6: Thanks for your comments about the self-healing mechanism of E-GES, and we are sorry for the lack of the interpretation of each mechanism in the manuscript.

We would like to explain the self-healing characteristics of E-GES from the following aspects. First of all, there are two kinds of self-healing mechanisms that polymer-based e-skin use to heal after suffering mechanical damage as reported by Bao et al⁹. The first one is to embed the microcapsules which contain self-healing agents inside the e-skin, and upon mechanical damage, these agents are released and induced polymerization at the damaged regions to achieve self-healing. The other one relies on the reformation of dynamic bonds on polymer chains to restore the original polymer properties. Such dynamic bonds include hydrogen bonds, metal-ligand coordination bonds, ionic interactions, disulfide bonds, and π - π interactions. **Obviously, the self-healing mechanism of E-GES belongs to the second one, and there are three dynamic bonds involved in self-healing process of E-GES, including hydrogen bonds, metal-ligand coordination bonds and disulfide bonds. Actually, endowing polymers with self-healing ability by the combination of these three types of dynamic chemical bonds has already been reported.** In this report, thioctic acid (TA) that contains dynamic disulfide bonds and hydrogen bonds was selected to react with iron (III) ions to prepare supramolecular polymers (Fig. R6), and the obtained product showed autonomous self-healing capability because of the existence of three types of dynamic chemical bonds, that is, dynamic covalent disulfide bonds, noncovalent H-bonds, and iron (III)-carboxylate coordinative bonds²³ (Fig. R7). Therefore, **we believe**

that the design principle of self-healing mechanism for E-GES is reasonable and effective.

Fig. R6 | The preparation of the poly (TA-DIB-Fe) copolymer network. This figure was taken from the article “Exploring a naturally tailored small molecule for stretchable, self-healing, and adhesive supramolecular polymers”²³.

Fig. R7 | The self-healing mechanism of poly (TA-DIB-Fe) copolymer network.

This figure was taken from the article “Exploring a naturally tailored small molecule for stretchable, self-healing, and adhesive supramolecular polymers”²³.

Subsequently, we would like to introduce the characteristics of gluten by illustrating dough-based electronics as examples. Basically, upon hydration and kneading, the gluten polymeric networks are built up through the formation of cross-linked disulfide bonds and intra- and intermolecular hydrogen bonds between different components of gluten matrix inside wheat dough^{11, 12, 24-28}. Therefore, due to the existence of disulfide bonds and hydrogen bonds in the gluten network of wheat dough, wheat dough has been developed to prepare flexible electronics with self-healing ability, such as flexible supercapacitor and e-skin^{11, 29}. In fact, the reason why we chose gluten as raw material for preparing e-skin is that with the presence of the gluten, the wheat dough prepared for making noodles can be stretched repeatedly and then recovered to the original shape, and the teared dough can be reunited through hand-kneading, so we think the inherent stretchability and healing characteristics make the gluten an ideal candidate for e-skin. **However, the mechanical strength of gluten network is too weak to prepare e-skin without the existence of starch granules that are another important component of the wheat flour, so there are several works for designing flexible electronics with wheat dough but without any reports about using the gluten as raw materials.** Paradoxically, the starch granules embedded in the gluten network prevent the further design of the gluten, so how to improve the gluten network structure for the e-skin preparation actually is a very challenging work. Similar to the

design ideas about TA mentioned before, we tailored the gluten network structure through the usage of EGaIn to form dynamic EGaIn-SH coordinative bonds, thus constructing extra chemical cross-linkings. Therefore, for the self-healing ability of E-GES, we have successfully constructed three types of dynamic chemical bonds, and these three chemical bonds had been proven to be effective in the self-healing process of e-skin. **We feel terribly sorry for missing the interpretation of established self-healing mechanisms of these chemical bonds and relevant references in previous manuscript. We have added more interpretation on this issue in the revised one.**

In this work, because the inherent self-healing ability of the gluten network has been reported in dough-based electronics^{11,29} and the self-healing mechanism based on liquid metal-sulfide interaction has also been proven in the liquid metal-embedded sulfur polymer³⁰, **we put the focus on how EGaIn influences the self-healing ability of gluten network.** The introduction of EGaIn induces the formation of EGaIn-SH coordinative bonds and improves the ability of tyrosine residues to form hydrogen bonds in gluten network. Based on the existence of these two dynamic chemical bonds and the inherent disulfide bonds, we can speculate that when two cut pieces of E-GES contacting, the dynamic reconstruction of these three dynamic chemical bonds in the contacting interfaces facilitates the initial self-healing process and then the disulfide dynamic covalent exchange further promotes the eventual recovery process³¹. **As for the contribution of each chemical bonds in the self-healing process**, normally all three chemical bonds undoubtedly contribute to form a relatively weak interface at the initial recovery stage, but the complete recovery mainly relies on disulfide bonds and

hydrogen bonds because these two chemical bonds are involved in the construction of gluten polymeric network framework, and obviously the number of them are far more than that of EGaIn-SH coordinative bonds. We have further modified the section about the description of the self-healing mechanism and provided more relevant references for readers to better understand the self-healing mechanism of E-GES.

Thank you for pointing out the inadequacies of the interpretation of the self-healing mechanism. We hope the revised version of the self-healing mechanism of E-GES is more understandable and clearer. Thank you very much.

ii) The authors noted the effect of the tyrosine side-chain of gluten for self-healing characteristics. However, the interpretation of the self-healing mechanism via tyrosine in E-GES is lacking. Therefore, the authors should provide the chemical structure with additional NMR data for the tyrosine group in E-GES. The authors should then supplement the additional schematics and explanations of the self-healing mechanism.

A6: Thanks for your comments and suggestions about the role of tyrosine (Tyr) in the self-healing mechanism of E-GES. We are sorry for the inadequate description of this part.

As for the established theories about gluten network, it is well accepted that Tyr residues that periodically appear in gluten proteins are involved in the construction of gluten network by forming hydrogen bonds³²⁻³⁴. As mentioned above, the dynamic hydrogen bonds play a significant role in the self-healing process, so revealing the influence of EGaIn toward the change of Tyr residues greatly help us to understand the

information of hydrogen bonds in the self-healing process. **The common research way for investigating the state of Tyr residues in proteins relies on the analyzation of Raman bands assigned to them.** Among these Raman bands, the doublets at around 850 and 830 cm^{-1} , arising from Fermi resonance of the ring-breathing vibration and an overtone of an out-of-plane ring-bending vibration of the para-substituted benzene ring, are most useful for monitoring the microenvironment around Tyr residues³⁵. **Two interpretations proposed for these bands** are listed below: The first is that the intensity ratio of these two bands depends on whether the Tyr residue is exposed or buried, while the other is that the ratio depends on the state of hydrogen bonding of the phenolic OH group. Some studies have pointed out that these two interpretations may be related, because the hydroxyl group of the exposed Tyr residue can interact with solvent and act as a simultaneous acceptor and donor of moderate to hydrogen bonds, while that of the buried Tyr residue tends to react with other amino acid residues as a hydrogen bond donor (Fig. R8)³⁴. **In the former situation, the intensity ratio $I_{850/830}$ tends to be high (0.9-1.45), whereas for the latter situation the ratio is lower (0.7-1.0).** Besides, the ratio can be as low as 0.3 in the case of extremely strong hydrogen bonding to a negative acceptor or as high as 2.5 for the case of a phenolic OH group, which acts as a strong hydrogen-bond acceptor. The detail discussion of these interpretations can be found in the work of Siamwiza et al. and *Protein Structure-Function Relationships in Foods*^{34, 35}.

Fig. R8 | Relative intensity of the 850 and 830 cm^{-1} doublets in the Raman spectrum for: (a) exposed tyrosine residues, which may act as hydrogen bond acceptor or donor; and (b) buried tyrosine residues, which act as hydrogen bond donor. This figure was taken from the book *Protein Structure-Function Relationships in Foods*³⁴.

As for E-GES, with the increasing content of EGaIn the intensity ratio increases from 1.17 to 1.65, meaning that Tyr residues tend to be exposed in gluten network and their phenolic oxygen serve primarily as an acceptor of protons. Therefore, it can be inferred that in the self-healing process of E-GES, **the exposed Tyr residues contribute to the formation of hydrogen bonds as proton acceptors or they can interact with each other to form intermolecular hydrogen bonds in cut interfaces**, suggesting that the addition of EGaIn benefits for the self-healing of the gluten network. We have supplemented additional explanations and schematics to further illustrate the self-healing mechanism via Tyr residues (Fig. R9). **Thank you for your opinions about the self-healing mechanism of our work. We believe that under your guidance, the improved version of the self-healing mechanism of E-GES can be understood more easily by readers.**

Fig. R9 | Schematic illustration of the formation of hydrogen bonds by the exposed Tyr residues.

However, we are very sorry that we are unable to get any valuable information of Tyr residues of gluten network from the obtained NMR data, as shown in Fig. R10. The prepared gluten network is insoluble in many solvents such as water, acetone, acetonitrile, tetrahydrofuran, dimethyl sulfoxide, toluene, methanol, chloroform and so on, and then the mixed solutions prepared by mixing different solvents also fail to dissolve it. Therefore, we can only choose the solid-state NMR instead of the liquid-state NMR for the detection of gluten network. However, there are no obvious differences between the data from the control sample and E-GES, and then the data cannot reflect all information about H in different chemical environment. **To this end, we can only use the already established method and theories mentioned above to analyze Tyr residues of gluten network.** We hope you can understand that. Thank you very much.

Fig. R10 | ¹H solid-state NMR of the control sample and 5% E-GES.

Revised section:

The self-healing mechanism based on reconstruction of dynamic bonds is widely accepted to be more preferable for e-skin applications¹. After introducing EGaIn into the gluten network to construct cross-linking bonds, there are three kinds of dynamic chemical bonds in E-GES, namely, S-S bonds, H-bonds and EGaIn-SH coordinative bonds. The self-healing mechanism of gluten networks based on reformation of dynamic S-S bonds and H-bonds has been accepted and reported in dough-based flexible electronics^{11,14}, and liquid metal-sulfide interactions have also been proven to be effective in the self-healing process of a liquid metal-embedded sulfur polymer⁴⁴. Therefore, the self-healing ability of E-GES can be inferred to mainly depend on the influence of the introduced EGaIn on S-S bonds and H-bonds in the gluten network. How EGaIn influences S-S bonds in E-GES has been discussed above based on the dynamic -SH/S-S rearrangement mechanism in the gluten network, so information on H-bonds in the E-GES network needs to be investigated. In the established theories about the gluten network, tyrosine (Tyr) residues that periodically appear in the gluten

proteins are involved in the construction of the gluten network by forming H-bonds, so information about H-bonds can be acquired by analysing the microenvironment of Tyr residues^{29,45,46}. Generally, the Tyr doublet in Raman spectra, arising from the Fermi resonance of the ring-breathing vibration and an overtone of an out-of-plane ring-bending vibration of the para-substituted benzene ring, is mostly used to analyse the state of Tyr residues (Fig. 3b)⁴⁷ because the variation in the intensity ratio of the Tyr doublet (I_{861}/I_{837}) provides H-bonding information of Tyr residues^{29,48}. An increase in the I_{861}/I_{837} ratio indicates that Tyr residues tend to be exposed to a hydrophilic environment and possess the ability to form strong hydrogen bonds, serving as proton acceptors, while a decreasing value means that Tyr residues are in a buried state. The ratio changing from 1.17 to 1.65 shows that the combination with EGaIn promotes Tyr residue exposure (Supplementary Fig. 9)⁴⁹, meaning that the exposed Tyr residues contribute to the formation of hydrogen bonds as proton acceptors or can interact with each other to form intermolecular hydrogen bonds (Supplementary Fig. 10)^{46,47}. Hence, based on the observations of Tyr residues of gluten chains and the presence of different dynamic chemical bonds in the gluten network, the proposed self-healing mechanism of E-GES can be summarized as follows: the introduction of EGaIn induces the formation of EGaIn-SH coordinative bonds and improves the ability of Tyr residues to form hydrogen bonds in the gluten network. Based on the existence of these two dynamic chemical bonds and the inherent disulfide bonds, when two cut pieces of E-GES contact each other, the dynamic reconstruction of these three dynamic chemical bonds in the contacting interfaces can be speculated to facilitate the initial self-healing

process. Meanwhile, the disulfide dynamic covalent exchange continually promotes the eventual recovery process, and the broken gluten network is finally rebuilt through the formation of cross-linked S-S bonds and intra- and intermolecular H-bonds (Fig. 3c). Similarly, a self-healing mechanism based on the combination of S-S bonds, H-bonds and metal-ligand bonds has also been reported in a thioctic acid-based supramolecular polymer¹⁶.

Supplementary Figure 10. Schematic illustration of the formation of hydrogen bonds by the exposed Tyr residues⁴.

Q7. Please check for some typo and figure errors. (Figure S7)

A7: Thanks for your comments. We are sorry for these careless mistakes and language problems. We have corrected these problem and we also use “Nature Communications proofreading services” to further edit our manuscript. We hope the revised manuscript can meet the requirements of Nature Communications. Thank you very much.

Revised figure:

Supplementary Figure 8. The loading-unloading cycle tests of 5% E-GES with varying maximum strains. The area of the stress-strain curves increases with the increase in the tensile strain, meaning that E-GES can dissipate more energies to withstand the strain change without breakage.

Finally, we sincerely appreciate your professional suggestions. From your opinions, we learn how to design experiments to show the sensing ability of E-GES. And you also kindly remind us the inadequate demonstration of some mechanism explanations, and provide us with relevant solving ideas. Thanks again for your guidance.

Reviewer #2 (Remarks to the Author):

Dear Editor & Authors

Overall, this work is an interesting study that possibly meets with the interest of the scientifically broad readership of Nature Communications. The result shows that relatively small amounts of Gallium and Indium can be used to actively control the crosslink density of the protein molecules due to the ability of the metals to interact and occupy the -SH thiol groups, creating metal ligands while reducing crosslink density and providing a softer material with stretchability. The story is relatively straightforward, and the novelty is high. No similar studies have been found in the literature.

The topic in itself seems timely selected in a more dominant science related to biopolymers and their investigated use as alternative resources on the market with more sceptics towards petroleum-based engineering plastics. It is believed that the set of experiments provided comprises in a relatively short format a good story to the readership, while at the same time displaying potentials with natural materials as such. It is in this sense inspiring reading that will probably receive follow-up work and citations.

Occasionally, it is difficult to distinguish between speculative statements and statements that the authors have based on their findings. This may be a consequence of technical proofreading carried out by someone that only have corrected the language but has not comprehended the message in what was said in the text. I have marked out

these sections so that the authors can address this problem, but it is still recommended that Nature Communications proofreading services review the text.

The list of comments provided, line by line, was made because the manuscript can be considered for publication in a revised format. The recommendation from my side, therefore, is a major revision is required.

Finally, the number of authors is 18! It is impossible to understand how it could have been necessary with several scientists to carry out simple IR verifications. The reviewer knows that the first three authors contributed equally, and the last two supervised the work. The rest of the authors need detailed motivation for their contribution since the total number of authors is almost twice as many as usual for these works in Nature Communications.

A: Thanks for your positive evaluation of our work.

(1) Since we determined to delve into this research topic, we have spent three years on this subject. During this period, we conducted countless experiments to explore the possibility to design protein-based e-skins and we also failed many times, but we continued to revise our experimental plans, as we believed that this work is meaningful.

We deeply appreciate your comments on our work, which makes us feel that our efforts have become worthwhile.

(2) We are sorry for some wrong English statements in our manuscript. We have tried our best to correct them firstly, and then used “**Nature Communications proofreading services**” to further improve our text. Besides, we are deeply grateful for your professional and detailed guidance for improving our manuscript. **We sincerely**

appreciate that we can communicate with you and we think that we obtain much enlightenment from your opinions. Thank you so much!

(3) Since our group lack the experience and detecting instruments to analyze protein structure, to ensure the accuracy of experimental results the characterization experiment in this work were divided into several parts and carried out by different research groups. They not only kindly used their own testing instruments to carry out characterization experiments, but also generously discussed experimental results with us. In addition to analyzing the obtained results with us, they also contributed to the interpretation of different mechanisms, such as the cross-linking mechanism, dynamic disulphied-sulphydryl exchange mechanism, self-healing mechanism and so on. Besides, they helped us modify the schematic illustrations of different mechanisms during the manuscript writing process. We deeply appreciate their valuable and profession suggestions for improving this work. **This work is the result of our joint efforts.** Therefore, to express our gratitude to them, we think it is our responsibility to list their names in the manuscript. We have further supplemented their contributions in the manuscript. Thanks for your understanding.

We have corrected the problems according to comments you listed one by one. **We hope the revised version of this manuscript can meet the publication requirements of “Nature Communications”. Thanks so much for your time and help.**

List of comments line by line:

1. Abstract:

Line 21 and 31: Biocompatibility? Allergenic people? What is meant?

A: Thanks very much for your detailed guidance about our manuscript. We would like to explain our understanding of biocompatibility from the following aspects.

As for the biocompatibility, since e-skin devices will interact closely with humans, they should be biocompatible, meaning that the raw materials for biocompatible e-skins should not negatively impact the host body and are also nature biocompatible⁹. At present, natural materials or nature-inspired materials, such as silk, cellulose, chitin, and so on, have been proven to be the unique options to endow e-skins with biocompatibility, because they are less likely to cause skin inflammation or other skin problems compared with most synthetic materials. Among those natural materials, silk fibroin (SF) has become one of the most popular candidates for e-skin preparation due to its natural biodegradability and biocompatibility. The obtained SF-based e-skins do show biocompatibility in the on-skin attachment tests and show biodegradability in the degradation tests^{8, 36}. Therefore, based on the reports of the established SF-based e-skins, we believe that natural proteins are attractive materials for preparing biocompatible e-skins, and thus we mention that “designing electronic skin (e-skin) with proteins is a critical way to endow e-skin with biocompatibility” in the manuscript. **In this work, the results from cytotoxicity tests and the designed on-skin attachment experiments prove that the prepared E-GES does not cause**

any adverse symptoms on the skin, so it can be concluded that E-GES is biocompatible when being attached to the skin.

It is an undeniable fact that 6-7% of the earth population has been estimated to suffer from gluten sensitivity, because they cannot digest the food containing gluten properly and thus the gluten intake will cause them to be sick. According to references^{37, 38} and an article of Dr Gwee Kok Ann who is the gastroenterologist practicing at Gleneagles Hospital, Singapore (“5 Gluten Sensitivity Myths Debunked”, <https://www.gleneagles.com.sg/healthplus/article/gluten-sensitivity-myths>), the gluten intolerance can be divided into different situations, including celiac disease, wheat allergy, gluten food intolerance, but these problem all originate from eating foods containing gluten. **Therefore, the contact of gluten is probably to be safe for these people, because in the reported allergenic cases, gluten exerts its effect as an ingestant rather than as a contactant.** We feel so sorry that we are not specialists in this field, so we cannot provide more data about the biocompatibility of E-GES toward people showing allergenic reactions to gluten.

However, according to the current definition of biocompatible e-skin reported by Bao et al, **we think E-GES basically meet the requirements of biocompatible e-skin due to its biocompatibility for skin and the natural biodegradability**⁹. Of course, to realize the practical application of E-GES, it should be further examined by more biocompatibility tests related to the allergenic people in the future. However, at least at this stage, we believe that gluten can be seen as a biocompatible material for the research of e-skin compared with most synthetic materials obtained from

petrochemicals. After all, although there are a certain number of people suffering from silk sensitivity in the world, silk still is the important raw material in the research of biocompatible e-skin³⁹. **Therefore, the above is our understanding to the biocompatibility of e-skin, and we hope the above explanation is reasonable and acceptable. Thank you very much.**

Line 27: Hierarchical Sulphur bonds... -What do the authors mean by hierarchical? If the authors mean that the disulfide bonds are subordinate in affecting mechanical properties (compared to the beta-sheets), this needs to be clarified.

A: Thanks for your comments about hierarchical sulphur bonds in abstract. After checking the abstract again, we think that the mention of “hierarchical sulphur bonds” may cause confusion to readers. What we want to express here is that in addition to the existence of S-S bonds in the obtained gluten network, there are EGaIn-SH coordinative bonds formed after the introduction of EGaIn. These two kinds of sulphur bonds have different functions in the gluten network, the former for maintaining the structure integrity⁴⁰ while the latter for improving the mechanical properties^{23, 41}, so we mention hierarchical sulphur bonds. However, due to the limitation of word count, without relevant explanations presented in abstract this concept easily confuses readers.

Therefore, to avoid confusion, we have removed these words and the meaning of the remaining sentence is not affected. Readers still can find detail explanations about different sulphur bonds in main text. Thank you very much.

Revised sentence:

The intrinsic reversible disulfide bond/sulfhydryl group reconfiguration of gluten networks is explored as a driving force to introduce EGaIn as a chemical cross-linker, thus inducing secondary structure rearrangement of gluten to form additional β -sheets as physical cross-linkers.

Line 30: To avoid confusion: “a synthetic natural like stretchability” is mentioned, but it confuses the reader to believe that stretchability refers to a synthetic (non-biobased) polymer material, instead of highlighting the stretchability integrated into the material as a property displaying stretch/ recovery values similar to, e.g., natural rubber. Also, see articles on natural rubber as sensors.

A: Thanks for your comment. We are so sorry for the ambiguity in this sentence due to the inaccurate description.

We mentioned “a synthetic material-like stretchability” in the abstract, because we want to express an idea that the protein-based e-skin also can achieve a high stretchability with appropriate adjustments. Currently, e-skins prepared by synthetic materials, such as poly(dimethylsiloxane), poly(vinylidene fluoride-co-hexafluoropropylene), poly(acrylic acid) and so on⁴²⁻⁴⁴, can easily realize high stretchability, while the stretchability of reported protein-based e-skins is fairly low. However, we prove that the stretchability of gluten protein-based e-skin can match that of e-skins designed based on synthetic materials in this work. Therefore, we want to highlight this characteristic in the abstract.

We have corrected this sentence in the abstract to make its meaning clearer.

Thank you very much for your careful guidance.

Revised sentence:

Remarkably, the obtained gluten-based material **is self-healing, achieves** synthetic material-like stretchability (>1600%) and **possesses the ability to promote** skin cell proliferation.

2. Introduction:

Line 40: outstanding mechanical properties deserve to be defined, e.g., as compared to what?

A: Thanks for your comments. We think that the use of “outstanding” to describe the mechanical properties is not accurate here after rethinking this sentence, and **“controllable” may be more suitable.** We want to express that in addition to skin-like sensing capabilities, the mechanical properties of e-skin also can be adjusted by different methods. We have replaced the word. Thank you very much.

Revised sentence:

The increasing demand for electronic skin (e-skin) in the fields of skin-attachable devices, robotics and prosthetics has motivated various cutting-edge technologies to endow e-skin with skin-like sensory capabilities and **controllable** mechanical properties, but unfortunately, e-skin with biocompatibility presents great challenges for practical on-skin applications.

Line 47: no need to refer to microstructure. Only structure is sufficient.

A: Thanks for your comments. We have revised this sentence accordingly.

Revised sentence:

However, the design of e-skin with proteins is still in its infancy because precisely controlling the **structure** of proteins to obtain adjustable mechanical properties and self-healing abilities is fairly complicated.

Line 52-54: protein is a polymer in itself. Simplify text to be more accurate and to the point.

A: Thanks for your comments. We have simplified the text and revised the sentence accordingly.

Revised sentence:

Upon hydration and kneading, gluten is known to form a cross-linked three-dimensional polymeric network through intra- and intermolecular covalent and noncovalent bonds

Line 57: 'loses,' is it meant 'misses'?; check with proofreading company.

A: Thanks for your comments. We are sorry for this mistake. It should be "lack" in this place. We have revised this sentence accordingly.

Revised sentence:

The gluten network possesses various dynamic bonds, such as dynamic covalent disulfide (S-S) bonds and noncovalent H-bonds, thus guaranteeing the self-healing

ability that most SF-based e-skins **lack**.

Line 60: Gluten is typically a hard/rigid network when no plasticizer is included in the consolidated material (e.g. H₂O), the statement about strengthening and toughening is awkward in the context of using EGaIn to break up the network. Only toughening likely applies here, needs better explanation otherwise, or reference, missing reference?

A: Thanks for your comments about this sentence. We are sorry that this statement is ambiguous and needs to be improved to express our thoughts more accurately. For synthetic materials, their mechanical properties, such as stiffness, tensile strength and toughness, can be adjusted by the formation of various cross-linking bonds, such as covalent bonds, hydrogen bonds and coordinative bonds in their network structure^{23, 42}. Similarly, the mechanical properties of SF-based materials also can be improved by introducing cross-linkers⁴⁵. At the same time, since β -sheet structures can be used as physical cross-linkers in the protein, achieving the adjustment of β -sheet structures is another effective method for improving the mechanical properties of protein-based materials. It has been reported that the tensile strength, toughness and stiffness of SF-based materials can be significantly enhanced by adjusting the β -sheet structures^{41, 46}.
⁴⁷.

However, the reports about enhancing the gluten network are rare, so we want to use chemical and physical cross-linkers together to improve the gluten network for e-skin preparation. **Because of the construction of different cross-linking positions, the stiffness, tensile strength and toughness of gluten network are improved**

simultaneously. Therefore, we think the statement about “strengthening and toughening” should be changed to “enhancing the mechanical properties”, which may make this sentence more accurate and to the point. We are sorry for the previous wrong expression and we have revised this sentence accordingly in the manuscript. Thank you very much.

Revised sentence:

However, **enhancing the mechanical properties** of the soft gluten network by achieving co-incorporation of physical and chemical cross-linking sites at the molecular level remains a challenge.

Line 65-75: rephrase the entire section for clarity. The “reversible -SH/S-S exchange mechanism” needs referencing and /or clarification. The -SH/S-S links are known to be possible to rearrange.

A: Thanks for your comments for this section. We are sorry that the description of this section is so complex that make it difficult for readers to understand, and we have rephrased this section in manuscript to make its meaning clearer.

Regarding the mention of “reversible -SH/S-S exchange mechanism”, because Stewart et al. revealed the disulphied-sulphydryl exchange in dough by incorporating ³⁵S-cysteine (³⁵S-CySH) into the gluten proteins of dough and found the following reaction equations (Fig. R11)³¹, the disulphied-sulphydryl exchange should be a dynamic reversible process when preparing gluten network. Therefore, we mentioned the “reversible -SH/S-S exchange mechanism” in the manuscript. However, **we think**

“rearrange” can be a better word to describe the disulphid-sulphydryl interchange, so we have rephrased this sentence and supplemented relevant references. Thanks a lot for your opinions!

Fig. R11 | Disulphide-sulphydryl exchange mechanism. These equations are taken from the article “DISULPHIDE-SULPHYDRYL EXCHANGE IN DOUGH”³¹.

Revised section:

To address this issue, the abundant free sulfhydryl (-SH) groups in the gluten network can be explored to construct cross-linking sites since they can be used as ligands to form cross-linking bonds via metal-ligand coordinative interactions. Based on this idea, the eutectic gallium indium alloy (EGaIn), a promising candidate for improving the mechanical properties of soft materials, has the potential to be introduced into the gluten network due to its ability to interact with thiolate ligands^{18,19}. Therefore, this work focuses on the structural characteristics of the gluten network to design an EGaIn/gluten-based e-skin (E-GES). Our strategy targets the dynamic -SH/S-S rearrangement mechanism in the gluten network²⁰ to introduce EGaIn as a chemical cross-linker and thus realize establishment of hierarchical S-bonds in E-GES in which S-S bonds maintain the structural integrity¹² while EGaIn-SH coordinative bonds are supposed to dissipate energy¹⁶. Surprisingly, this structural adjustment strategy can

induce changes in the gluten backbone conformation to obtain additional β -sheets as physical cross-linkers^{21,22}.

Line 72: s-s bonds maintain the structural integrity while EGaIn -sh coordination bonds contribute to dissipating energy. This should be referenced or clear to the reader that it is a speculative proposed theory of the authors.

A: Thanks for your comments about this sentence. We have supplemented relevant references in corresponding sentence for readers to better understand the functions of these two chemical bonds. Because it is generally accepted that S-S bonds play a significant role in the formation and functionalities of three-dimensional gluten network⁴⁰, the construction of gluten network in E-GES should be attributed to S-S bonds and thus we mentioned “S-S bonds maintain the structural integrity” in manuscript. Besides, achieving the reversible cross-linking of polymer chains is an effective way to implement mechanical energy dissipation⁴¹ and metal-ligands coordinative bonds also have been proven to dissipate energy²³, so the formation of coordinative bonds between EGaIn and sulfhydryl groups of gluten chains are supposed to possess the ability to dissipate energy. We have supplemented references for this statement.

Line 75: The authors mean an increase in Beta -sheet contents.

A: Thanks for your comments. We have corrected this error.

Line 88-97: The referencing should appropriately highlight the previous research on gluten materials, referencing representative research over the last 30 years instead of referencing a few works since 2014, since this knowledge was known long before.

A: Thanks for your comments. We have supplemented some representative references in the manuscript for readers to get more knowledges of gluten materials. We sincerely apologize for that.

Line 85: “for developing”

A: We are sorry for this grammatical error and have corrected it. Thank you very much.

Revised sentence:

In summary, through the dynamic network microregulation mechanism of gluten networks, we realize a combination of liquid metal and protein to achieve protein-based e-skin, which could provide insights into metal-protein interactive mechanics **for developing** more proteins for e-skin.

Line 97: How do the authors confirm no starch? Check materials section. No purifications steps were described. At least the protein contents should be provided in the materials section.

A: Thanks for your comments. Because the starch embedded in the gluten network will influence the direct design for gluten network, we use gluten as raw material instead of wheat flour to prepare E-GES. **We noticed that the CAS number of glutens is 8002-80-0, so to ensure the consistency of gluten in each experiment we choose to**

purchase the gluten from Meryer (Shanghai) Chemical Technology Co., Ltd instead of washing the wheat dough to remove the starch and thus obtaining gluten. We think that the gluten that is made into a product for sale should be purified before, so its purity should be guaranteed. After investigation, we found that the initial supplier of the gluten that we bought is TCI (Shanghai) DEVELOPMENT CO., LTD., and **the MSDS report of gluten that we got from this company shows that the gluten is not a mixture** (Fig. R12). Therefore, we mentioned “there are no starch granules embedded in the gluten network of E-GES” in the manuscript.

*This SDS for user in China - Not correspond to the regulation of other regions.

TCI (Shanghai) DEVELOPMENT CO., LTD.

Revision number: 1 Revision date: 09/06/2019 Page 1 of 4

Revision date: 09/06/2019

SAFETY DATA SHEET

1. IDENTIFICATION

Product name:	Gluten from Wheat
Product code:	G0066
Company:	TCI (Shanghai) DEVELOPMENT CO., LTD.
Address:	No.96 Pu Gong Road, Shanghai Chemical Industry Park, Shanghai 201507 China
Responsible Department:	Sales Department
Telephone:	+86-(0)21-67121386
Fax:	+86-(0)21-67121385
e-mail:	Sales-CN@TCIchemicals.com
Emergency telephone:	0532-83889090
Revision number:	1

2. HAZARDS IDENTIFICATION

Classification of the substance or mixture	
PHYSICAL HAZARDS	Not classified
HEALTH HAZARDS	Not classified
ENVIRONMENTAL HAZARDS	Not classified
Label elements	
Pictograms or hazard symbols	None
Signal word	No signal word
Hazard statements	None
Precautionary statements	None

3. COMPOSITION/INFORMATION ON INGREDIENTS

Substance/mixture:	Substance
Components:	Gluten from Wheat
Percent:
CAS RN:	8002-80-0
Chemical Formula:	----

Fig. R12 | The excerpt of MSDS report of gluten obtained from TCI (Shanghai) DEVELOPMENT CO., LTD.

Line 100: small mass? NaCl? 1 wt.%, 3 wt.% etc

A: We are sorry for this colloquial expression. We have supplemented specific values in this place.

Revised sentence:

As shown in Fig. 1a, E-GES was prepared by mixing gluten with EGaIn-dispersed solutions with the addition of a small amount of NaCl (2.8 wt.% E-GES) and kneading.

Line 114: The text in Fig 1b and c are too small to see properly. Central information in Fig 1 should be enlarged. Non-important popular scientific imaging can be made smaller.

A: Thanks for your comments. We have readjusted the scale and position of the images in Fig 1. We are sorry for the previous composition of Fig 1.

Revised figure:

Line 115: ‘support these observations’ – better English

A: Thanks for your comments. We have corrected it and revised this sentence accordingly.

Revised sentence:

The increasing trend of free -SH content, released from the combination with EGaIn during the analytical test, **supports these observations**.

Line 116: It is unclear what Fig 1d-g refers to, the legend of the entire Fig 1 needs revision for improved clarity.

A: Thanks for your comments. After proving that EGaIn has been introduced in the gluten network through the interaction with -SH groups of gluten, we think **whether the integrity of gluten network will be influenced by the introduction of EGaIn is**

a critical problem. Therefore, field emission scanning electron microscopy (FESEM) was used to observe the micromorphology of E-GES. SEM micrographs show that the microstructure of E-GES (Fig.1e) is similar to that of the control sample (Fig.1d), and more importantly the gluten network microstructure becomes flatter and denser than before, rather than loosely layered, meaning that the formation of EGaIn-SH coordinative bonds is benefit to the improvement of gluten network. Besides, the energy-dispersive X-ray spectroscopy (EDS) mapping results reveal the existence of EGaIn in gluten network (Fig. 1f, g), so based on those experimental results listed in the manuscript and EDS mapping results we can prove that we have successfully introduced into the gluten network. **We are sorry for the unclear meaning and we have revised the relevant sentence and the legend of Fig 1. Thank you very much.**

Revised sentence:

(1) Furthermore, **to ensure the integrity of the gluten network after introducing EGaIn, field emission scanning electron microscopy (FESEM) was used to observe the E-GES microstructure.** Fig. 1d, e show that **the gluten network of E-GES is more regular and denser than that of the control sample, rather than loosely layered,** meaning that the decrease in the S-S content does not influence the structural integrity of the gluten network. In addition, the energy-dispersive X-ray spectroscopy (EDS) mapping result reveals the presence of EGaIn in E-GES (Supplementary Fig. 3). Therefore, EGaIn has been successfully introduced into the gluten network through the construction of intermolecular EGaIn-SH coordinative bonds and in turn contributes to the adjustment of the gluten network.

(2) **Fig. 1 | Schematic drawings of preparing E-GES and analysis of the interaction between EGaIn and -SH groups of gluten in E-GES.** **a**, Schematic illustration of the E-GES fabrication process. **b**, The XPS spectra of the control sample (top) and 5% E-GES (bottom). **-SH groups are effective ligands for coordination with EGaIn, and S-Ga bonding can be revealed by XPS.** For the control sample, the S 2p region shows the S 2p^{3/2} and S 2p^{1/2} components at binding energies of 163.8 eV and 164.9 eV, respectively, while these two peaks in the 5% E-GES sample show shifts in the low binding energy direction **due to the change in electron density of S**, appearing at 161.6 eV and 164.5 eV, respectively. This indicates the formation of EGaIn-SH coordinative bonds in E-GES, consistent with previous reports^{16,26}. **c**, Free -SH and S-S contents of different E-GES samples. **In the E-GES preparation process, with increasing amount of EGaIn, more -SH groups tend to interact with EGaIn to form coordinative bonds, thus reducing their ability to form S-S bonds with each other. Therefore, as the EGaIn content increases, the S-S content decreases, and a higher -SH content can be detected.** **d, e**, SEM micrographs of the control sample (**d**) and the 5% E-GES sample (**e**). **Observations of the microstructure of the E-GES gluten network demonstrate that although the addition of EGaIn causes a decrease in the S-S content, the gluten network structure remains intact, meaning that the formation of EGaIn-SH coordinative bonds contributes to enhancing the gluten network.** **f**, Photograph of a stretched E-GES sample. The 5% E-GES sample can be easily stretched more than 10 times. **g**, Photograph of E-GES with different shapes. E-GES can be moulded into different complex shapes, i.e.,

knot, bird and giraffe, **which is beneficial to the design of different shapes for irregular human skin.**

Line 127-129: redundant - remove

A: Thanks for your comments. We have deleted this redundant sentence.

Line 130-132: the values should be given some comparison, for example: How do these values compare to commercial polymer for skin coverage such as PDMS patches of silicone?

A: Thanks for your comments. We are sorry that our original writing idea was not rigorous enough and we recognized that **the lack of sufficient data comparison would seriously affect the readers' comprehensive understanding of the mechanical properties of E-GES after receiving your professional opinions.** We deeply appreciate your valuable comments.

We have supplemented the mechanical properties of the commonly commercial used e-skin substrates, including poly(dimethylsiloxane) (PDMS) (sylgard 184), polyurethane (SG80A) and styrene-ethylene-butadiene-styrene (SEBS), to compare with that of E-GES in the manuscript⁴².

Supplementary content:

For commonly commercialized e-skin substrates, such as poly(dimethylsiloxane) (PDMS) (Sylgard 184), polyurethane (SG80A) and styrene-ethylene-butadiene-styrene (SEBS), their stretchability is lower than that of E-GES, exhibiting breaking strains of

200%, 700% and 280%, respectively, but they are stiffer than E-GES, with Young's moduli of 0.4 MPa, 1.73 MPa and 3.83 MPa, respectively⁴². However, the Young's moduli of these commercial synthetic materials are often greater than the maximum value of the skin's Young's modulus (600 kPa, the mechanical properties of skin in different parts of the body differ)³². Compared with these materials, the softer and more stretchable E-GES may become a better choice for preparing e-skin.

Line 135-136: ...minimum value of 1200%, see fig 2a for 10 wt.%EGaIN.?

A: Thanks for your comments. We have corrected this careless mistake. We are so sorry for that.

Revised sentence:

In contrast, the breaking strain exhibits a decreasing trend and shows a minimum value of approximately 1200%.

Line 145-146: ...alfa-helix in all EGES samples...: here is needed clarification because, at this point, the reader has not been introduced to alfa-helix molecular conformations. The % of alfa-helix molecular conformations is mentioned and compared for previous works, but no value of the % is provided for the E-GES. Why? High content? ...or is it only that poor English was used and the content was determined to 20-30%? As it stands now, these values are related to previous studies.

A: Thanks for your comments. We are sorry that the description of alfa-helix in this section is insufficient, which will cause confusions to readers. Moreover, the logic of

this sentence is not clear enough, so readers cannot correctly understand the meaning conveyed by authors. We have supplemented the introduction of alfa-helix for readers to know about this conformation firstly, and listed specific value of alfa-helix contents to discuss the influence of its content changes on gluten network. The listed “20-30%” values are related to previous studies, and we are sorry for this unclear sentence. We have rephrased this sentence to make the comparison between this work and previous works clearer. Thanks a lot for your opinions.

Supplementary content:

According to previous studies, the change in α -helix content can indicate the integrity of the protein network because an increasing content is related to the construction of a more ordered protein network structure, while a decreasing content suggest the breakage of intermolecular S-S bonds and rearrangement of protein molecules^{38,39}. The addition of EGaIn causes the α -helix content in E-GES to decrease, which may result from the rearrangement of -SH/S-S induced by EGaIn and thus the reduction of intermolecular S-S bonds; however, all E-GES samples have more than 33% α -helix structures in their gluten network (Supplementary Fig. 6). The α -helix contents were approximately 20% to 30% in previous gluten modification research, so E-GES actually has a more ordered gluten network structure^{12,13}.

Line 148-155: In this section, the authors need to introduce the reader to the difference in conformation between alfa helices and the beta sheets and provide relevant references on these matters. Not every reader is familiar with these conformational

differences, and it is also necessary since the authors in the next section present thermal stability results (line 157-159). Alfa helices and beta sheets are only one part of the entire protein macromolecular complex. What about the tertiary and quaternary protein structures?

A: Thanks for your opinions. We have supplemented the introduction of beta sheets in the manuscript for readers to distinguish the function of beta sheet from that of alfa-helix in gluten network. Moreover, we also provided relevant references about the basic knowledge of these two conformations structure and about the detailed mechanism of using FTIR to analyze the secondary structure of proteins. We hope that through this way readers can have a clear idea about the secondary structures of proteins.

Besides, we think that the description of thermal stability results is not good enough, and the logic of this section also has some problems, so we have rephrased the whole section to discuss the thermal stability of E-GES and supplemented more relevant references for readers to understand the results. We hope that the meaning of this new section is clear enough and readers can understand it easily.

Because we want to enhance the mechanical properties of gluten network through the construction of cross-linking sites and we know that the crystalline β -sheets can act as the physical cross-linkers in the protein network, we focus on revealing how the introduction of EGaIn influences the β -sheets in gluten network to explore the gluten network enhancing mechanism. The addition of EGaIn in the gluten network do cause the reduction of S-S, which is an important factor in discussing the change of the tertiary structure. Generally, to determine the changes of tertiary structure in gluten network,

the tested gluten samples usually are dissolved in phosphate buffer solution firstly, and then fluorescence spectrum, ultraviolet absorption spectrum and circular dichroism spectrum can be used to obtain the information of gluten tertiary structure. However, the construction of different cross-linking sites further improves the gluten network of E-GES, so the obtained E-GES samples are difficult to dissolve in phosphate buffer solution, causing the difficulty in detecting the tertiary structure of E-GES samples through the general way. Although the above three testing technologies can be used to test solid samples, relevant reports about analyzing and discussing the tertiary structure of gluten samples are rare. Therefore, it is a great pity that based on the reported references it is difficult for us to organize a clear research idea about how to study the tertiary structure of E-GES. Besides, there is almost no research and references on the quaternary structures of gluten. Therefore, maybe at present the research theories and methods on the tertiary structure and quaternary structure of gluten are not yet fully developed, compared with the established studies of its secondary structures.

Moreover, the strategy provided in this work for improving the gluten network is new and there are no similar references reported before, so to ensure the accuracy and reliability of experimental results we need to clearly reveal the changes in the secondary structure of E-GES firstly. And then based on the established results, we can further design new methods and/or cooperate with other research groups to explore the tertiary and quaternary structures of E-GES.

As for the gluten network enhancing mechanism studied in this work, the analysis results of secondary structure of E-GES can clearly show the influence of EGaIn to

gluten network at this stage, and the obtained results are sufficient to support the feasibility of the applied network regulation strategy. Therefore, we want to report the experimental achievements of this strategy firstly, and then continue to further explore the gluten network change mechanism of E-GES. **We deeply appreciate your advice about the tertiary and quaternary protein structures, and we think you are the best reader of our work. Thank you very much.**

Supplementary content:

(1) The changes in the gluten backbone conformation were revealed by analysing amide I bands ($1700\text{-}1600\text{ cm}^{-1}$), **and the deconvolution of amide I bands can be used to qualitatively and quantitatively evaluate gluten secondary structures**, as illustrated in Fig. 2d, e and Supplementary Fig. 5^{12,13,37}.

(2) **In addition, the β -sheet is referred to as the most stable conformation of gluten, and its content is positively related to the viscoelasticity and rigidity of the gluten network¹².**

The incorporation of EGaIn facilitates the formation of a higher proportion of β -sheets at the expense of α -helices in a dose-dependent manner. This result is likely attributed to the reduction of intermolecular S-S bonds and the **promotion** of interactions among intramolecular H-bonds induced by EGaIn⁴⁰, **so EGaIn facilitates the construction of a more rigid and highly viscoelastic gluten network in E-GES. Meanwhile**, notably, these nanocrystal-like β -sheets can act as many physical cross-linking points embedded in the amorphous protein network, thus defining the mechanical properties of the protein, as widely found in studies on silk protein^{21,40}.

Line 157-159: “The enhancing thermal stability...” : Here, the authors need to correct their statement to proper English (although the reviewer can see what the authors aim at stating from a scientific results perspective).

A: Thanks for your comments. We have rephased the whole section about the thermal stability of E-GES samples. We are sorry for the original unclear statements. We hope that the new section is clear enough for readers to understand the meaning of thermal stability experiments in our work. Thank you for your guidance.

Supplementary content:

Generally, when protein chains are cross-linked, their mobility is limited, causing an increase in the denaturation temperature (T_p) of the protein. Thus, T_p is an important indicator of changes in protein chains^{22,41}. Differential scanning calorimetry (DSC) was used to detect the T_p of different E-GES samples (Fig. 2f). T_p increases from 48.26 °C to 58.17 °C under different EGaIn dosages, indicating that the mobility of gluten protein chains is restricted with the addition of EGaIn, and thus, the gluten network spatial structure becomes stronger and more ordered, instead of tending to becoming denatured and weaker (Supplementary Table 1)³⁸. Moreover, in addition to the β -sheet physical cross-linking sites, there are abundant chemical cross-linking points because of the coordinative interactions between EGaIn and the -SH groups of gluten, so the adjustment of the E-GES gluten network should be attributed to the synergistic promotion of these two cross-linking mechanisms²².

Line 160-162: At this point in the text, the authors again refer to polymer physics and how thermal stability can be influenced for specific structures due to more rigid arrangements / stricter molecular conformations, highlighting the need for brief info to the reader about the structures (above). Additional and better referencing should be provided.

A: Thanks for your comments. We think the mention of polymer physics is not appropriate in this place, so we have reconstructed the writing logic of this section and put the explanation about how to analyze the cross-linked protein chains at the beginning of this section. Besides, compared with the mention of polymer physics, we think we should provide relevant references about the influences of cross-linked gluten chains to gluten network structure, which may be more helpful for readers to understand the detailed mechanism behind it, so we have deleted the sentence about polymer physics. With better references provided in this section, we hope the modification of this section are qualified to be presented to readers. Thanks for your professional suggestions.

Line 165-166: rephrase: "... crosslinking sites synergistically promote the adjustment of the EGeS network structure". The meaning of this is unclear.

A: Thanks for your comments. We have rephrased this sentence to make the logic of this section more coherent and make the meaning of the sentence clearer. We are sorry for the unclear description of this sentence. Thank you very much.

Revised sentence:

Moreover, in addition to the β -sheet physical cross-linking sites, there are abundant chemical cross-linking points because of the coordinative interactions between EGaIn and the -SH groups of gluten, so the adjustment of the E-GES gluten network should be attributed to the synergistic promotion of these two cross-linking mechanisms.

Line 169-174: proper referencing to recent and most informative articles on how rheological behaviour vary for semisolid protein hydrogels in relation to the presence of metal ions is required. The authors currently only relate mechanical properties change to crosslinks and beta sheets, whereas in reality, 10% metal ions have been introduced. See reference: doi.org/10.1021/acsnano.0c10893

A: Thanks for your comments. We named E-GES samples according to the mass ratio of EGaIn in gluten, so when using 6g of gluten to prepare 10% E-GES, 0.6g of EGaIn is required and EGaIn is added to the gluten in the form of EGaIn-dispersed solution. Because the mass ratio of gluten to EGaIn-dispersed solution is 1:1, the quality of EGaIn-dispersed solution is 6g when the gluten is 6g. Therefore, the proportion of EGaIn in the obtained 10% E-GES is 5%. And according to the relative molecular mass of EGaIn, the concentration of EGaIn in 10% E-GES is around 7.4 mM. Therefore, the concentration of EGaIn in 10% E-GES is fairly low compared with the concentrations of metal ions used for protein nanofibrils and their hydrogel formation, so we think **the key factor affecting the rheological properties of E-GES still is the change of the gluten network structure**. We are sorry that there are no relevant references about the influence of metal ions to the rheological behavior of protein hydrogels provided in this

section. Thank you for sharing this reference. We think this work clearly reveals the relation between the valence state and ionic radius of metal ions and the formation of protein nanofibrils hydrogel, and we learn a lot from this work. We have supplemented this reference in the manuscript. Thank you very much.

Line 171: English problem: The improving... It should state: The enhanced... The use of a proofreading agency that actually reads what is being said in the article is necessary. It is clear that the work has been through professional proofreading but not with a language critical person who has understood the meaning of the sentences (sometimes).

A: Thanks for your comments. We have corrected this problem and we are sorry for many English problems in this manuscript.

Revised sentence:

The **enhanced** viscoelasticity of E-GES agrees well with its enhanced mechanical performance in tensile tests.

Line 177: remove the word *microscopic*. It has no meaning here.

A: Thanks for your comments. We have deleted this word.

Line 188-190: This statement needs to be proven and shown with a photo.

A: Thanks for your comments. Actually, Fig. 2i show photographs of the inflation process of E-GES by the air pump, but we made a mistake that we introduced Fig. 2i to the previous sentence. We apologize for that. We have corrected this problem. Thank

you very much.

Revised sentence:

As shown in Fig. 2i, the membrane-shaped E-GES can be inflated into a balloon with a size range several orders of magnitude larger than before, which is an amazing phenomenon for protein-based materials or hydrogel-like materials.

Line 203-206: The section needs clarification because it is not possible to understand the specifics. What is a circular wound?

A: Thanks for your opinions. We are sorry for the wrong description of this sentence.

We have rewritten this sentence. Thank you very much.

Revised sentence:

To examine the ultimate self-healing property of E-GES during the tensile experiment, two cut-off E-GES pieces were combined for 1 min without any further treatment; under this seriously restricted healing condition, the contact interfaces cannot fully disappear in the healed E-GES.

Line 210: remove poor English, 'is deeply', write: was

A: Thanks for your comments. We have corrected this mistake.

Line 214: what is 'ratio valve'?

A: Thanks for your comments. We have corrected this description. We are sorry for that.

Revised sentence:

An increase in the I₈₆₁/I₈₃₇ ratio indicates that Tyr residues tend to be exposed to a hydrophilic environment and possess the ability to form strong hydrogen bonds, serving as proton acceptors, while a decreasing value means that Tyr residues are in a buried state.

Line 217-226: the authors need to be clear about that they speculate on the mechanism and that their suggested mechanism is based on the xx and yy observations. Are the authors talking about gluten protein denaturation in the presence of EGaIn? Have the authors reflected over which one of the gluten protein fractions (e.g. gliadin or glutenin) that dominantly is exposed in the self-healing mechanism? Gliadin and Glutenin have very different amino acid compositions.

A: Thanks for your comments. We have corrected the description about the proposed self-healing mechanism in the text. Because of the existence of gluten network, wheat dough can be regarded as a material with self-healing ability¹¹. Therefore, how the introduction of EGaIn influences the self-healing ability of gluten network is the research focus in this section. Generally, it has been accepted that the construction of dynamic chemical bonds, including hydrogen bonds, metal-ligand coordination bonds, ionic interactions, disulfide bonds and π - π interactions, is an effective method to endow e-skin with self-healing ability⁹. For E-GES, there are hydrogen bonds, metal-ligand

coordination bonds and disulfide bonds in its gluten network. EGaIn-SH coordinative bonds and the disulfide bonds have been discussed in previous sections, so we mainly discuss the influence of the introduction of EGaIn to hydrogen bonds through analyzing Tyr residues of gluten chains in this section. **In summary, we want to discuss the self-healing mechanism of E-GES from the perspective of dynamic chemical bonds.** According to the experimental results, it can be inferred that the exposed Tyr residues are benefit to the formation of hydrogen bonds in cut interfaces after suffering cutting, thus contributing to the self-healing of gluten network.

Although the exposed Tyr residues may indicate the denaturation of gluten protein to some extent, results from FTIR and DSC tests have proven the stability of the gluten network of E-GES. Generally, protein denaturation results in the molecular structure transforming from an ordered to disordered state, from a folded to an unfold state, and from a natural to denatured state. These changes will be accompanied by an energy change. When protein hydrogen bonds are ruptured, protein molecules will unfold and absorb heat. Therefore, T_p decreases, suggesting that the gluten network spatial structure has changed and became weaker and more disordered, and the gluten protein is more prone to denaturation⁴⁸. However, the T_p of E-GES increases, meaning that the gluten network of E-GES is not denatured. Besides, according to the analysis results of secondary structures of E-GES, the stable conformation structures occupy the dominant proportion in gluten network. Based on the discussion mentioned above, we think the gluten network of E-GES is not denatured but becomes stronger than before. **Therefore,**

it can be inferred that the existence of some exposed Tyr residues does not affect the stability of gluten network structure.

Besides, according to previous reports, glutenin proteins consist of two subunit populations: high molecular weight glutenin subunits (HMW-GS) which have apparent molecular weights in the range of 80,000-120,000 Da and low molecular weight glutenin subunits (LMW-GS) with weights of 40,000-55,000 Da. Gliadin proteins are monomeric and classified as ω -, α/β -, and γ -gliadins based on their mobility in electrophoretic gel systems and their amino acid sequences⁴⁹. The HMW-GS and LMW-GS together form a disulfide crosslinked protein polymeric network through the formation of intra- and intermolecular disulfide bonds thereby contributing to the strength and the elastic properties of the gluten. Gliadins have only intramolecular disulfide linkages and do not contribute to the protein matrix formation, but they interact with the glutenin structures via non-covalent bonds and thus affect the viscous properties of the gluten network²⁴. **Therefore, regarding the contribution of different gluten protein fractions in the self-healing mechanism of E-GES, we infer that glutenin proteins contribute more than gliadins proteins due to their ability to construct the gluten network through intra- and intermolecular disulfide bonds.** However, we feel so sorry that our research group lack relevant knowledge and technologies to study different protein fractions and their amino acid compositions, so we cannot provide experimental results to support our speculation. We are so sorry for that.

Therefore, at the current research stage, we decided to analyze the dynamic chemical bonds in the gluten network of E-GES to demonstrate its self-healing mechanism, which also is the dominant research idea for revealing the self-healing mechanism of e-skin in current reports⁹. We are sorry that we cannot provide a more detailed interpretation of the self-healing mechanism against the characteristics of gluten protein at this stage. Thanks for your understanding. **We will try our best to dissolve these problems on our ongoing research programs.**

Supplementary content:

(1) The self-healing mechanism based on reconstruction of dynamic bonds is widely accepted to be more preferable for e-skin applications¹. After introducing EGaIn into the gluten network to construct cross-linking bonds, there are three kinds of dynamic chemical bonds in E-GES, namely, S-S bonds, H-bonds and EGaIn-SH coordinative bonds. The self-healing mechanism of gluten networks based on reformation of dynamic S-S bonds and H-bonds has been accepted and reported in dough-based flexible electronics^{11,14}, and liquid metal-sulfide interactions have also been proven to be effective in the self-healing process of a liquid metal-embedded sulfur polymer⁴⁴. Therefore, the self-healing ability of E-GES can be inferred to mainly depend on the influence of the introduced EGaIn on S-S bonds and H-bonds in the gluten network. How EGaIn influences S-S bonds in E-GES has been discussed above based on the dynamic -SH/S-S rearrangement mechanism in the gluten network, so information on H-bonds in the E-GES network needs to be investigated. In the established theories about the gluten network, tyrosine (Tyr) residues that periodically appear in the gluten

proteins are involved in the construction of the gluten network by forming H-bonds, so information about H-bonds can be acquired by analysing the microenvironment of Tyr residues^{29,45,46}. Generally, the Tyr doublet in Raman spectra, arising from the Fermi resonance of the ring-breathing vibration and an overtone of an out-of-plane ring-bending vibration of the para-substituted benzene ring, is mostly used to analyse the state of Tyr residues (Fig. 3b)⁴⁷ because the variation in the intensity ratio of the Tyr doublet (I_{861}/I_{837}) provides H-bonding information of Tyr residues^{29,48}.

(2) The ratio changing from 1.17 to 1.65 shows that the combination with EGaIn promotes Tyr residue exposure (Supplementary Fig. 9)⁴⁹, meaning that the exposed Tyr residues contribute to the formation of hydrogen bonds as proton acceptors or can interact with each other to form intermolecular hydrogen bonds (Supplementary Fig. 10)^{46,47}. Hence, based on the observations of Tyr residues of gluten chains and the presence of different dynamic chemical bonds in the gluten network, the proposed self-healing mechanism of E-GES can be summarized as follows: the introduction of EGaIn induces the formation of EGaIn-SH coordinative bonds and improves the ability of Tyr residues to form hydrogen bonds in the gluten network. Based on the existence of these two dynamic chemical bonds and the inherent disulfide bonds, when two cut pieces of E-GES contact each other, the dynamic reconstruction of these three dynamic chemical bonds in the contacting interfaces can be speculated to facilitate the initial self-healing process. Meanwhile, the disulfide dynamic covalent exchange continually promotes the eventual recovery process, and the broken gluten network is finally rebuilt through the formation of cross-linked S-S bonds and intra- and intermolecular H-bonds (Fig. 3c).

Similarly, a self-healing mechanism based on the combination of S-S bonds, H-bonds and metal-ligand bonds has also been reported in a thioctic acid-based supramolecular polymer¹⁶.

Line 228: in a great measure = to a great extent (English)

A: Thanks for your comments. We have corrected this problem.

Revised sentence:

Intriguingly, we find that ultralow temperatures cannot cause irreversible damage to E-GES because it can recover its mechanical and conductive properties to a great extent as the temperature rises.

Line 231: status should read experiments

A: Thanks for your comments. We have replaced the wrong word.

Revised sentence:

The strain sensitivity of E-GES was systematically evaluated based on the resistance changes caused by different stretching-releasing experiments.

Line 234: indicating the electrical stability as a consequence of elasticity of the polymer network, it should read.

A: Thanks for your guidance. We have corrected this sentence.

Revised sentence:

As shown in Fig. 3f, the resistance of E-GES first increases when stretched and then decreases after release, and no obvious conductivity hysteresis is observed, **indicating the electrical stability as a consequence of the elasticity of the polymer network.**

Line 236: again, the English need revision from someone skilled in technical writing, e.g. 'after suffering from complete cutting'

A: Thanks for your comments. We feel so sorry for our English problems. Thank you very much for your detailed guidance.

Revised sentence:

Moreover, **after healing from complete cutting**, the healed E-GES is capable of restoring its original resistance (Fig. 3h).

Line 243 to 285 requires a reviewer knowledgeable in the fields of biocompatibility. From this reviewer's perspective, it is unclear how these materials are used with people showing allergenic reactions towards gluten protein. 6-7% of the earth population has been estimated to suffer from gluten sensitivity.

A: Thanks for your comments about the biocompatibility tests. After reviewing the literatures on gluten sensitivity^{37, 38}, we find that for people with gluten intolerance or sensitivity, they may experience pain and bloating after eating foods that contain gluten. Research indicates that around 1% of people in the United States have celiac disease, 1% have a wheat allergy, and 6% or more have gluten intolerance (The above data is from the website article "What is gluten intolerance?",

<https://www.medicalnewstoday.com/articles/312898>). According to the article of Dr Gwee Kok Ann who is the gastroenterologist practicing at Gleneagles Hospital, Singapore, “celiac disease is an autoimmune disorder where ingestion of gluten causes damage to the intestinal lining, generating an immune response that could cause inflammatory reactions in many parts of the body; gluten sensitivity is a digestive disorder where a person reacts aversely to gluten, usually with symptoms of the gastrointestinal tract such as diarrhoea, bloating or gas; in the large majority of gluten food intolerance it is to the poorly digested, and easily fermentable, constituents of these foods and thus with these foods, the majority of people will develop symptoms of bloating and gas if a large quantity is consumed. Many times when a person feels discomfort after eating certain foods, it is intolerance, not an allergy. Food intolerance is generally harmless, and can be managed by just reducing the total amount eaten at one time, and taking them in small quantities” (The above excerpts are from the article of Dr Gwee Kok Ann, “5 Gluten Sensitivity Myths Debunked”, <https://www.gleneagles.com.sg/healthplus/article/gluten-sensitivity-myths>).

Therefore, we think that the stagnation of gluten in intestines and stomach is the main reason for gluten allergic reactions. According to the inducement mechanism of gluten intolerance or sensitivity, the in vitro use of gluten as strain sensor should be safe for people showing allergenic reactions towards gluten, because the fabricated gluten network cannot enter the body through the skin when being attached to skin. Based on the discussions above, **we think that evaluating whether E-GES is**

a biocompatible e-skin should still rely on the biocompatibility tests against the skin.

Moreover, it should be noted that the biocompatibility of SF-based e-skins is mainly demonstrated by the results of the on-skin attachment experiments in previous reports^{8, 36, 50}. However, **these experiments were further improved in our work**. We attached E-GES not only to the healthy skin, but also to the skin with a #-shaped wound. The obtained results demonstrated that E-GES did not cause skin adverse reactions. Besides, according to the professional suggestions from Professor Guilliat, we also supplemented the cytotoxicity tests against the human epidermal keratinocytes cell lines and human dermal fibroblasts cell lines to evaluate the cytocompatibility of E-GES. Therefore, we think the biocompatibility tests of E-GES against the skin in our manuscript are fairly complete and comprehensive, and the obtained results successfully demonstrate that E-GES is a biocompatible e-skin. We are very grateful for your opinions about the biocompatibility of E-GES, and we hope that the above explanations are reasonable and convincing. Thank you very much.

Line 286-309:

What are authors reflections about similar works on strains sensors such as this work, <https://www.nature.com/articles/ncomms4132>

and how can the authors relate their work to previous works on strain sensors? Overall, the referencing in the manuscript is rather shallow.

A: Thanks for your comments. We are sorry for missing relevant references in this section. For the work provided in your comments, Gong et al, used gold nanowire-impregnated tissue paper and poly-dimethylsiloxane sheets to prepare a sensitive pressure sensor which can resolve pressing, bending, torsional forces and acoustic vibrations and can realize real-time monitoring of blood pulses as well as the detection of small vibration forces from music. Compared with this multifunction sensor, it is a pity that E-GES cannot sense the pressure signals and its function is relatively single.

Currently, through some ingenious strategies, e-skins prepared by synthetic materials or SF have achieved the integration of different sensing abilities, such as the combination of pressure and strain sensing abilities. As for the most reported PDMS-based e-skins, it has been reported that one of them can be used to detect strain, electrocardiogram and heart rate information⁴. Moreover, a reported SF-based e-skin possesses the ability to acquire strain, humidity and temperature signals⁵¹. **In contrast, the sensing ability of E-GES is relatively single function and needs to be further developed.** We think this is the main problem of E-GES, and we have supplemented this reflection in the manuscript.

Besides, we also supplemented the comparison of strain sensing ability between E-GES and other reported e-skins to discuss the strain sensing features of E-GES. We hope the revised version of this section can give readers a clearer idea about the strain sensing ability of E-GES. We are sorry for the previous version. We deeply appreciate your suggestions for improving our manuscript.

Supplementary content:

(1) Therefore, compared with the established SF-based e-skins^{6,7,50}, E-GES can not only sense strain changes from human motions of different scales but also be used as a built-in sensor integrated in a mask to detect the slight changes induced by breathing, suggesting that E-GES is an attractive e-skin for strain sensing and that gluten is another potential protein for e-skin preparation in addition to SF.

(2) Therefore, in addition to sensing strain signals of human motions, E-GES possesses the potential to be used in robotics, in which e-skin with high stretchability in three dimensions will enable robots to perform more complicated motions, rather than simply imitating human motions⁹.

(3) Hence, E-GES is robust and has the ability to resist mechanical damage in daily use, comparable to the robustness of e-skins prepared from synthetic materials, such as PDMS and poly(acrylic acid)^{35,51}. However, the e-skins designed with synthetic materials or SF have achieved integration of different sensing abilities. For example, PDMS-based e-skin has been reported to have both strain and electrocardiogram sensing capabilities at the same time⁵¹, and SF-based e-skin can be used as humidity or temperature sensors⁸. In contrast, the sensing ability of E-GES is relatively single function and needs to be further developed. Therefore, utilizing gluten to design e-skins with multifunctional sensing abilities is still a challenging task. Achieving regulation of tertiary and quaternary structures of gluten may be a key point in developing various sensing abilities of gluten-based e-skins.

Line 311: Have the authors created a new protein?

A: Thanks for your comments. We are sorry for this wrong description. We have corrected it and revised the sentence accordingly.

Revised sentence:

In summary, we report a **simple strategy to design gluten protein** for fabricating e-skin with stretchability, self-healing ability and biocompatibility.

At last, we are very grateful for your opinions. We deeply appreciate that you have provided us with many detailed and professional suggestions. When answering your question, we found many mistakes that we had overlooked during the writing process and we think we must correct these mistakes. We have learned a lot from your detailed guidance, such as how to construct the writing logic clearly, how to express our ideas to readers, and how to cite references more correctly. Meanwhile, we sincerely apologize for our English writing problems. We are very honored to be able to communicate with you. In this process, we have gained a lot. Once again, sincerely thank you for your time and help.

References

1. Hammock ML, Chortos A, Tee BC, Tok JB, Bao Z. 25th anniversary article: The evolution of electronic skin (e-skin): a brief history, design considerations, and recent progress. *Adv. Mater.* **25**, 5997-6038 (2013).
2. Miyamoto A, et al. Inflammation-free, gas-permeable, lightweight, stretchable on-skin electronics with nanomeshes. *Nat. Nanotechnol.* **12**, 907-913 (2017).
3. Li B, et al. Adaptable ionic liquid-containing supramolecular hydrogel with multiple sensations at subzero temperatures. *J. Mater. Chem. C* **9**, 1044-1050 (2021).
4. Son D, et al. An integrated self-healable electronic skin system fabricated via dynamic reconstruction of a nanostructured conducting network. *Nat. Nanotechnol.* **13**, 1057-1065 (2018).
5. Liang J, et al. Intrinsically stretchable and transparent thin-film transistors based on printable silver nanowires, carbon nanotubes and an elastomeric dielectric. *Nat. Commun.* **6**, 7647 (2015).
6. Wang Q, Jian M, Wang C, Zhang Y. Carbonized Silk Nanofiber Membrane for Transparent and Sensitive Electronic Skin. *Adv. Funct. Mater.* **27**, 1605657 (2017).
7. Chen G, et al. Plasticizing Silk Protein for On-Skin Stretchable Electrodes. *Adv. Mater.* **30**, 1800129 (2018).
8. Hou C, et al. A Biodegradable and Stretchable Protein-Based Sensor as Artificial Electronic Skin for Human Motion Detection. *Small* **15**, 1805084 (2019).
9. Yang JC, Mun J, Kwon SY, Park S, Bao Z, Park S. Electronic Skin: Recent Progress and Future Prospects for Skin-Attachable Devices for Health Monitoring, Robotics, and Prosthetics. *Adv. Mater.* **31**, 1904765 (2019).
10. Zhou Y, et al. Highly Stretchable, Elastic, and Ionic Conductive Hydrogel for Artificial Soft Electronics. *Adv. Funct. Mater.* **29**, 1806220 (2019).
11. Lei Z, Huang J, Wu P. Traditional Dough in the Era of Internet of Things: Edible, Renewable, and Reconfigurable Skin-Like Iontronics. *Adv. Funct. Mater.* **30**, 1908018 (2019).
12. McCann TH, Day L. Effect of sodium chloride on gluten network formation, dough microstructure and rheology in relation to breadmaking. *J. Cereal Sci.* **57**, 444-452 (2013).
13. Galal AM, Varriano-Marston E, Johnson JA. Rheological dough properties as affected by organic acids and salt. *Cereal Chem.* **55**, 683-691 (1978).

14. Tuhumury HC, Small DM, Day L. Effects of Hofmeister salt series on gluten network formation: Part I. Cation series. *Food Chem.* **212**, 789-797 (2016).
15. Tuhumury HC, Small DM, Day L. Effects of Hofmeister salt series on gluten network formation: Part II. Anion series. *Food Chem.* **212**, 798-806 (2016).
16. Kinsella JE, Hale ML. Hydrophobic associations and gluten consistency: effect of specific anions. *J. Agr. Food Chem.* **32**, 1054-1056 (1984).
17. Miller RA HR. Role of Salt in Baking. *Cereal Foods World* **53**, 4-6 (2008).
18. Song Y, Liu Y, Qi T, Li GL. Towards Dynamic but Supertough Healable Polymers through Biomimetic Hierarchical Hydrogen-Bonding Interactions. *Angew. Chem., Int. Ed. Engl.* **57**, 13838-13842 (2018).
19. Xu Y, et al. White-light-emitting flexible display devices based on double network hydrogels crosslinked by YAG:Ce phosphors. *J. Mater. Chem. C* **8**, 247-252 (2020).
20. Liu M, et al. Endowing recyclability to anti-adhesion materials via designing physically crosslinked polyurethane. *J. Mater. Chem. A* **7**, 22903-22911 (2019).
21. Gao Y, Gu S, Jia F, Gao G. A skin-matchable, recyclable and biofriendly strain sensor based on a hydrolyzed keratin-containing hydrogel. *J. Mater. Chem. A* **8**, 24175-24183 (2020).
22. Fang G, et al. Exploration of the tight structural-mechanical relationship in mulberry and non-mulberry silkworm silks. *J. Mater. Chem. B* **4**, 4337-4347 (2016).
23. Zhang Q, Shi C-Y, Qu D-H, Long Y-T, Feringa BL, Tian H. Exploring a naturally tailored small molecule for stretchable, self-healing, and adhesive supramolecular polymers. *Sci. Adv.* **4**, eaat8192 (2018).
24. Tuhumury HCD, Small DM, Day L. The effect of sodium chloride on gluten network formation and rheology. *J. Cereal Sci.* **60**, 229-237 (2014).
25. Khan K, Shewry PR, Khan K, Shewry PR. *Wheat: chemistry and technology*. (Wheat : chemistry and technology 2009).
26. Belton PS. On the elasticity of wheat gluten. *J. Cereal Sci.* **29**, 103-107 (1999).
27. Shewry PR, Halford NG, Belton PS, Tatham AS. The structure and properties of gluten: an elastic protein from wheat grain. *Phil. Trans. R. Soc. Lond. B* **357**, 133-142 (2002).
28. Macritchie F. Physicochemical Properties of Wheat Proteins in Relation to Functionality. In: *Advances in Food and Nutrition Research* (ed Kinsella JE). (Academic Press 1992).

29. Hu M, et al. A flour-based one-stop supercapacitor with intrinsic self-healability and stretchability after self-healing and biodegradability. *Energy Storage Mater.* **21**, 174-179 (2019).
30. Xin Y, Peng H, Xu J, Zhang J. Ultrauniform Embedded Liquid Metal in Sulfur Polymers for Recyclable, Conductive, and Self-Healable Materials. *Adv. Funct. Mater.* **29**, 1808989 (2019).
31. Mauritzen CAM, Stewart P. Disulphide—Sulphydryl Exchange in Dough. *Nature* **197**, 48-49 (1963).
32. Wieser H. Chemistry of gluten proteins. *Food Microbiol.* **24**, 115-119 (2007).
33. Nawrocka A, Szymańska-Chargot M, Miś A, Wilczewska AZ, Markiewicz KH. Effect of dietary fibre polysaccharides on structure and thermal properties of gluten proteins – A study on gluten dough with application of FT-Raman spectroscopy, TGA and DSC. *Food Hydrocolloids* **69**, 410-421 (2017).
34. Yada R, Jackman R, Smith J. *Protein Structure-Function Relationships in Foods*. (Springer, Boston, MA 1994).
35. Mwindaae, et al. Interpretation of the doublet at 850 and 830 cm⁻¹ in the Raman spectra of tyrosyl residues in proteins and certain model compounds. *Biochemistry* **14**, 4870-4876 (1975).
36. Huang J, et al. Stretchable and Heat-Resistant Protein-Based Electronic Skin for Human Thermoregulation. *Adv. Funct. Mater.* **30**, 1910547 (2020).
37. Barbaro MR, Cremon C, Stanghellini V, Barbara G. Recent advances in understanding non-celiac gluten sensitivity. *F1000Research* **7**, F1000 Faculty Rev-1631 (2018).
38. Khalid AN, McMains KC. Gluten sensitivity: fact or fashion statement? *Curr. Opin. Otolaryngo.* **24**, (2016).
39. Pecquet C. Allergic reactions to insect secretions. *Eur. J. Dermatol. Ejd* **23**, 767-773 (2013).
40. Du J, Dang M, Khalifa I, Du X, Xu Y, Li C. Persimmon tannin changes the properties and the morphology of wheat gluten by altering the cross-linking, and the secondary structure in a dose-dependent manner. *Food Res. Int.* **137**, 109536 (2020).
41. Balu R, et al. Tough Photocrosslinked Silk Fibroin/Graphene Oxide Nanocomposite Hydrogels. *Langmuir* **34**, 9238-9251 (2018).
42. Kang J, et al. Tough and Water-Insensitive Self-Healing Elastomer for Robust Electronic Skin. *Adv. Mater.* **30**, 1706846 (2018).

43. Cao Y, et al. Self-healing electronic skins for aquatic environments. *Nat. Electron.* **2**, 75-82 (2019).
44. Darabi MA, et al. Skin-Inspired Multifunctional Autonomic-Intrinsic Conductive Self-Healing Hydrogels with Pressure Sensitivity, Stretchability, and 3D Printability. *Adv. Mater.* **29**, 1700533 (2017).
45. Choi J, Hasturk O, Mu X, Sahoo JK, Kaplan DL. Silk Hydrogels with Controllable Formation of Dityrosine, 3,4-Dihydroxyphenylalanine, and 3,4-Dihydroxyphenylalanine-Fe(3+) Complexes through Chitosan Particle-Assisted Fenton Reactions. *Biomacromolecules* **22**, 773-787 (2021).
46. Su D, Yao M, Liu J, Zhong Y, Chen X, Shao Z. Enhancing Mechanical Properties of Silk Fibroin Hydrogel through Restricting the Growth of beta-Sheet Domains. *ACS Appl. Mater. Interfaces* **9**, 17489-17498 (2017).
47. Tsuchiya K, Masunaga H, Numata K. Tensile Reinforcement of Silk Films by the Addition of Telechelic-Type Polyalanine. *Biomacromolecules* **18**, 1002-1009 (2017).
48. Huang L, Zhang X, Zhang H, Wang J. Interactions between dietary fiber and ferulic acid changed the aggregation of gluten in a whole wheat model system. *Lwt* **91**, 55-62 (2018).
49. Shewry PR, Halford NG, Belton PS, Tatham AS. The structure and properties of gluten: an elastic protein from wheat grain. *Phil. Trans. R. Soc. Lond. B* **357**, 133-142 (2002).
50. Seo J-W, Kim H, Kim K, Choi SQ, Lee HJ. Calcium-Modified Silk as a Biocompatible and Strong Adhesive for Epidermal Electronics. *Adv. Funct. Mater.* **28**, 1800802 (2018).
51. Wang Q, Ling S, Liang X, Wang H, Lu H, Zhang Y. Self-Healable Multifunctional Electronic Tattoos Based on Silk and Graphene. *Adv. Funct. Mater.* **29**, 1808695 (2019).

REVIEWERS' COMMENTS

Reviewer #1 (Remarks to the Author):

The authors have tried to address all issues raised by us in the revised manuscript. Obviously, they added a couple of experimental interpretation as well as clarification on sensing mechanism as for liquid metal-tailored gluten network based e-skin. Also, they not only revised some typo and figure errors, but also supported reliability characteristics.

However, I suggest that they should explain their priority in terms of self-healing mechanism and human motion sensing capability, compared to previous works more critically. Still, I would like to point out that they should modify this manuscript for the possible readers to better and more clearly understand their work in the field of e-skin.

Reviewer #2 (Remarks to the Author):

All questions raised by the reviewers have been carefully handled.

The point-by-point responses to reviewers' comments

Reviewer #1 (Remarks to the Author):

The authors have tried to address all issues raised by us in the revised manuscript. Obviously, they added a couple of experimental interpretation as well as clarification on sensing mechanism as for liquid metal-tailored gluten network based e-skin. Also, they not only revised some typo and figure errors, but also supported reliability characteristics.

However, I suggest that they should explain their priority in terms of self-healing mechanism and human motion sensing capability, compared to previous works more critically. Still, I would like to point out that they should modify this manuscript for the possible readers to better and more clearly understand their work in the field of e-skin.

A: Thank you very much for your affirmation and approval of our revised manuscript, and thanks again for your detailed guidance, which makes the idea of this article clearer and the content more complete.

We know that fabricating e-skin with gluten protein is a bold attempt and the idea of this work may be foreign to most researchers in the field of e-skin. However, we believe that using proteins to prepare e-skin is an interesting strategy to endow e-skin with skin-like sensing abilities considering that the normal sensory functions of skin are closely related to the cooperation of different proteins. Meanwhile, we feel so sorry that our previous manuscript may be not perfectly convey this idea to possible readers.

We have supplemented the priority of E-GES in terms of self-healing mechanism and human motion sensing capabilities in the manuscript accordingly. We hope that the modified manuscript can make possible readers better and more clearly understand our work. We deeply appreciate your professional opinions and we are pleasure to communicate with you.

Supplementary content:

(1) In this work, targeting a current limited number of proteins and corresponding structural design strategy suitable for protein-based e-skin preparation, we successfully show that gluten is an attractive alternative to SF in the field of e-skin. Here, through the dynamic network microregulation mechanism of gluten networks, we realize a combination of liquid metal and protein to achieve gluten-based e-skin, based on whose dynamic bonds analysis, a reliable self-healing mechanism for fabricating self-healing protein-based e-skins is suggested. Besides, the gluten-based e-skin performs well in human motion strain sensing, comparable to synthetic material-based e-skins. This work can provide insights into metal-protein interactive mechanics for developing more proteins for designing human skin-like e-skins. (These contents have been supplemented in the Introduction section, Page 5.)

(2) Obviously, the above discussion has successfully shown that the self-healing principle based on the reconstruction of dynamic bonds also applies to protein-based e-skin, suggesting that exploring the dynamic covalent bonding of proteins or constructing additional dynamic bonds inside protein structure is a great strategy to

endow protein-based e-skins with self-healing ability. (These contents have been supplemented in the Results section, Page 14.)

(3) Notably, hydrogel-like E-GES is more convenient to prepare strain sensors, with neither the introduction of elastomeric polymer substrates nor the complex preliminary treatment, compared with reported SF-based e-skins^{5, 7}. (These contents have been supplemented in the Results section, Page 16.)

(4) From the above experiment results, gluten protein can be inferred to have the potential to be an attractive alternative to synthetic materials for designing e-skin with more skin-like strain sensitivity or to be a new elastic polymer substrate in the field of e-skin. (These contents have been supplemented in the Results section, Page 20.)

(5) This work provides an attractive idea to design e-skin through the construction of protein networks by liquid metal, clearly demonstrates the self-healing mechanism based on the use of dynamic bonds also suitable for endowing protein-based e-skins with self-healing ability and successfully shows the vast potential of protein-based e-skin in sensing human motion. This method could be further developed by using other proteins, such as gelatin and egg albumin, and other inorganic materials, such as MXenes, carbon nanotubes and silver nanofibres, to obtain an e-skin with desired functions. (These contents have been supplemented in the Discussion section, Page 21.)

Reviewer #2 (Remarks to the Author):

All questions raised by the reviewers have been carefully handled.

A: Thank you very much for your appreciation of our carefully revised manuscript. We are extremely excited to receive your positive comments and we are immensely grateful for your help. Thanks a lot.